# A genome-wide association study based on the China Kadoorie Biobank identifies genetic associations between snoring and cardiometabolic traits
Yunqing Zhu[1], Zhenhuang Zhuang[1], Jun Lv[1,2,3], Dianjianyi Sun [1,2,3], Pei Pei[2], Ling Yang [4,5], Iona Y. Millwood [4,5], Robin G. Walters [4,5], Yiping Chen [4,5], Huaidong Du [4,5], Fang Liu[6], Rebecca Stevens[5], Junshi Chen[7], Zhengming Chen [5], Liming Li [1,2,3] & Canqing Yu [1,2,3] ✉ On behalf of the China Kadoorie Biobank Collaborative Group*

Despite the high prevalence of snoring in Asia, little is known about the genetic etiology of snoring and its causal relationships with cardiometabolic traits. Based on 100,626 Chinese individuals, a genome-wide association study on snoring was conducted. Four novel loci were identified for snoring traits mapped on *SLC25A21*, the intergenic region of *WDR11* and *FGFR*, *NAA25*, *ALDH2*, and *VTI1A*, respectively. The novel loci highlighted the roles of structural abnormality of the upper airway and craniofacial region and dysfunction of metabolic and transport systems in the development of snoring. In the two-sample bi-directional Mendelian randomization analysis, higher body mass index, weight, and elevated blood pressure were causal for snoring, and a reverse causal effect was observed between snoring and diastolic blood pressure. Altogether, our results revealed the possible etiology of snoring in China and indicated that managing cardiometabolic health was essential to snoring prevention, and hypertension should be considered among snorers.

Snoring is the repetitive episodes of complete or partial upper airway obstruction during sleep, which creates noise when breathing[1]. Habitual snoring is more severe, caused mainly by organic disorders[2]. Cohabitants could easily detect snoring due to the bothering noise. In China, habitual snoring was higher in males (13.6–31.4%) than in females (4.3–16.7%)[3]. Habitual snoring was probably related to cardiometabolic diseases, such as type 2 diabetes[4], making it a public health problem. Besides, snoring is a common symptom of obstructive sleep apnea. Considering the difficulties and costs of obstructive sleep apnea treatments, it's important to screen and prevent snoring[2].

Twins and family studies reported the heritability of snoring varying from 18–28%[5,6]. A genome-wide association study (GWAS) of the UK Biobank (UKB) has identified 41 genome-wide significant loci[7]. In contrast, the genetic basis of snoring for Asian people is still unknown.

Cardiometabolic factors might also be related to the snoring problem. Previous Mendelian randomization (MR) analysis conducted in Europeans reported bi-directional relationships between higher body mass index (BMI), diastolic blood pressure (DBP), and snoring. However, this hasn't been confirmed in Asians. Furthermore, no MR study has investigated the causal associations with other cardiometabolic traits, such as smoking and lipid metabolites.

The present study conducted a large-scale GWAS analysis based on the China Kadoorie Biobank (CKB) with two aims (1) to identify genome-wide significant loci of snoring among Asians; (2) to explore the genetic correlations and associations between snoring and cardiometabolic traits.

[1]Department of Epidemiology & Biostatistics, School of Public Health, Peking University, Beijing 100191, China. [2]Peking University Center for Public Health and Epidemic Preparedness & Response, Beijing 100191, China. [3]Key Laboratory of Epidemiology of Major Diseases (Peking University), Ministry of Education, Beijing 100191, China. [4]Medical Research Council Population Health Research Unit at the University of Oxford, Oxford OX3 7LF, United Kingdom. [5]Clinical Trial Service Unit & Epidemiological Studies Unit (CTSU), Nuffield Department of Population Health, University of Oxford, Oxford OX3 7LF, United Kingdom. [6]Suzhou Centers for Disease Control, NO.72 Sanxiang Road, Gusu District, Suzhou 215004 Jiangsu, China. [7]China National Center for Food Safety Risk Assessment, Beijing 100022, China. *A list of authors and their affiliations appears at the end of the paper. ✉e-mail: yucanqing@pku.edu.cn

We determined four novel loci of snoring and habitual snoring, which were mapped on *SLC25A21*, the intergenic region of *WDR11* and *FGFR*, *NAA25*, *ALDH2*, and *VTI1A*, respectively. The novel genes indicated that structural abnormality of the upper airway and craniofacial region, and dysfunction of the transportation system played important roles in the development of snoring. Post-GWAS analysis showed that snoring genes overlapped with obesity gene sets. Through the MR analysis, we found that general obesity and blood pressure were associated with a higher risk of snoring, and snoring was reversely associated with a higher level of DBP. The causal associations demonstrated that maintaining cardiometabolic health was essential for the prevention and treatment of snoring, and snoring could be an indicator of preventing hypertension among East Asians.

## Results

### GWAS for snoring, habitual snoring
Among CKB participants ($n = 100,626$), 46.9% were snorers, including 22,985 (22.8%) habitual snorers. Snoring frequency differed in the ten study areas (Supplementary Data 1). 55.0% of males and 40.9% of females were snorers. The habitual snorers were more likely to be elders, males, with geographical origins in southern China, not the genetic outliers within each geographical origin, with higher BMI, WC, and blood pressure, and more likely to be weekly drinkers and current smokers (all $P < 0.05$) (Supplementary Data 2). A strong correlation was found between snoring and sex (OR [95%CI] for snoring=1.784 [1.738-1.830]). Also, a correlation was found between snoring and BMI (OR = 1.162 [1.157–1.167] [per 1 kg m$^{-2}$]).

Four loci for snoring were identified in the primary GWAS, including one significant locus ($P < 5 \times 10^{-9}$), and the other three loci were suggestively significant ($P < 5 \times 10^{-8}$). Besides, two loci for snoring were identified in each of the BMI-adjusted, male-specific, and meta-analysis of study-area level GWAS analyses. The primary GWAS of habitual snoring identified three loci, all of which were significant. Both BMI-adjusted and meta-analysis of study-area level GWASs identified three loci for habitual snoring.

Comparing with the GWAS catalog results, we determined four novel loci for the snoring traits (including two significant loci). The novel locus mapped on *SLC25A21* was identified in the primary GWAS of both snoring traits. Additionally, a novel locus mapped on the intergenic region of *WDR11* and *FGFR* was identified in the BMI-adjusted GWAS analysis of both snoring traits. The snoring GWAS among the males identified two novel loci mapped on *NAA25*, *ALDH2*, and *VTI1A* genes.

Other loci were reported to be associated with snoring in the previous GWAS of UKB. Single nucleotide polymorphisms (SNPs) on the *FTO*, *BDNF-AS* and *BDNF* genes provided the most significant effect for primary GWASs of snoring ($P = 1.60 \times 10^{-9}$) and habitual snoring ($P = 1.20 \times 10^{-9}$), respectively. Besides, loci identified for snoring (or habitual snoring) were also associated with habitual snoring (or snoring) ($P < 1 \times 10^{-5}$) (Table 1, Fig. 1, Supplementary Data 3, 4, Supplementary Fig. 1). Linkage disequilibrium score (LDSC, version 1.0.1) estimates of SNP-based heritability were 10.5% (standard error = 0.89%) for snoring and 16.9% (standard error = 1.24%) for habitual snoring.

### Post-GWAS analysis
For snoring, rs2277339 identified in the primary GWAS was in the exonic region of the *PRIM1* gene, and rs671 in linkage disequilibrium (LD) with rs116873087 (identified in the male-specific GWAS) was in the exonic region of the *ALDH2* gene. For habitual snoring, rs6265 in LD with rs140138951 (identified in the primary GWAS) was in the exonic region of the *BDNF* gene (Supplementary Data 5).

Through expression quantitative trait loci (eQTL) mapping, five and three protein-coding genes for snoring and habitual snoring were additionally identified. *FTO*, *SLC25A21*, *RP11-964E11.2*, and *PAX9* were shared between the two snoring traits in the primary GWAS. The latter three genes were newly identified in CKB. Moreover, they were also mapped as novel genes for BMI-adjusted snoring traits. Significant SNPs were associated with the expression of genes in several tissues, including skeletal muscle, cells, skin, and brain. Heatmaps showed that snoring and habitual snoring genes were primarily expressed in the brain, skin, and metabolic tissues. *FTO*, *SLC25A21*, and *PRIM1* were identified by both positional and eQTL mapping (Supplementary Fig. 2, Supplementary Data 6-8).

Furthermore, hypergeometric tests were used for the gene-set enrichment analysis. Habitual snoring genes were overrepresented in the obesity gene sets. *BDNF*, *BDNF-AS*, and *FTO* were overrepresented in the BMI gene set ($P_{FDR} = 3.80 \times 10^{-6}$, FDR meant false discovery rate) and obesity gene set ($P_{FDR} = 2.69 \times 10^{-5}$). *FTO* and *IRX3*, the prioritized genes of habitual snoring, were overrepresented in the *FTO*-obesity-variant-mechanism gene set ($P_{FDR} = 0.0005$). Besides, prioritized snoring genes were overrepresented in the trunk fat mass gene set ($P_{FDR} = 0.039$), and habitual snoring genes were overrepresented in the coronary artery disease gene set ($P_{FDR} = 0.005$) (Fig. 2, Supplementary Data 9).

### Bidirectional replication
A total of 11 genomic risk loci identified in the primary and sensitivity analyses were included in the replication analysis. All significant loci identified in the primary analysis passed replication in UKB (all $P$-values in UKB $< 5 \times 10^{-5}$). One and two SNPs identified in GWAS of BMI-adjusted-snoring and habitual snoring, all the SNPs identified in GWAS for ten study areas, passed the replication with UKB GWAS summary statistics (all $P$-values in UKB $< 5 \times 10^{-5}$). The trans-ancestry minor allele frequency (MAF) comparison showed that most snoring loci had higher MAF in the East Asian population of CKB than in the European population of UKB ($P = 0.0336$) (Supplementary Data 10, Supplementary Fig. 3a). Especially, the nonreplicated loci were likely due to a relatively low allele frequency (<0.03) among the European population of UKB (Supplementary Data 11, Fig. 3a).

In the replication for UKB loci, 35 SNPs passed the quality control (QC) in CKB GWAS and were included in the present analysis. Among them, 14 SNPs (40%) passed the replication in CKB (all $P$-values in CKB < 0.05), five SNPs had the reverse direction, and others had $P$-value > 0.05 in CKB (Supplementary Data 12, Fig. 3b). The MAF difference between the East Asian population of CKB and the European population of UKB was more pronounced among the SNPs identified in the UKB population ($P = 0.0009$) (Supplementary Data 10, Supplementary Fig. 3b).

### PRS for snoring
The median interval between the baseline and the second resurvey was 8.0 years (interquartile range: 7.4–8.6). Among the independent target sample of snoring ($n = 17,951$), 46.9% snored at baseline, 49.5% snored at the second resurvey, the corresponding prevalences of habitual snoring were 28.3% and 30.6% among the target sample of habitual snoring ($n = 11,494$).

SNPs from CKB summary statistics ($P$ for threshold <0.390) were included in the best polygenic risk scores (PRSs) for snoring at baseline ($R^2_{PRS} = 0.0066$, $P_{PRS} = 2.01 \times 10^{-20}$). Participants in the highest snoring PRS decile had a 1.67 (95%CI:1.45–1.91) folds probability of snoring at baseline compared with those in the lowest decile. The best PRS of resurvey snoring was from UKB ($R^2_{PRS} = 0.0081$, $P_{PRS} = 6.25 \times 10^{-25}$), and the PRSs for habitual snoring at baseline ($R^2_{PRS} = 0.0158$, $P_{PRS} = 9.38 \times 10^{-28}$) and resurvey were from CKB ($R^2_{PRS} = 0.0128$, $P_{PRS} = 1.46 \times 10^{-23}$) (Supplementary Fig. 4, Supplementary Data 13, 14).

### Genetic correlations
Genetic correlation analyses indicated shared links between snoring traits and five cardiometabolic traits among Asians. Positive genetic correlations between snoring with BMI ($r_g = 0.39$, $P_{FDR} = 1.63 \times 10^{-21}$), body weight ($r_g = 0.27$, $P_{FDR} = 1.15 \times 10^{-10}$), systolic blood pressure (SBP) ($r_g = 0.16$, $P_{FDR} = 0.0081$), and ever smoked ($r_g = 0.14$, $P_{FDR} = 0.025$) were observed. Similar results were observed for habitual snoring, except for a significant correlation with DBP ($r_g = 0.14$, $P_{FDR} = 0.049$) (Fig. 4, Supplementary Data 15). Among the Europeans, snoring was positively correlated with adiposity-related traits, ever smoked, levels of glucose, glycated hemoglobin, triglycerides, blood pressure, and was negatively correlated with height, high-density lipoprotein cholesterol (Supplementary Data 16).

**Table 1 | Novel loci associated with snoring and habitual snoring in CKB**

| SNP | Chromosome | Position | A1 | A1FREQ | Info | *P* | β | SE | Positional mapped genes (eQTL mapped genes) |
|---|---|---|---|---|---|---|---|---|---|
| Snoring | | | | | | | | | |
| rs712398[a] | 14 | 37385687 | C | 0.594 | 1.000 | 1.40E-08 | 0.050 | 0.009 | *SLC25A21 (PAX9, SLC25A21, RP11-964E11.2)* |
| BMI-adjusted-snoring | | | | | | | | | |
| rs10886864 | 10 | 122929537 | C | 0.395 | 0.982 | 2.00E-10 | -0.056 | 0.009 | *WDR11:FGFR2* |
| rs712398 | 14 | 37385687 | C | 0.594 | 1.000 | 2.40E-10 | 0.055 | 0.009 | *SLC25A21 (PAX9, SLC25A21, RP11-964E11.2)* |
| Snoring in male | | | | | | | | | |
| rs116873087[a] | 12 | 112511913 | G | 0.793 | 0.966 | 1.10E-08 | 0.098 | 0.017 | *NAA25, ALDH2* |
| rs12265047[a] | 10 | 114487925 | G | 0.283 | 0.987 | 4.40E-08 | -0.083 | 0.015 | *VTI1A* |
| Habitual snoring | | | | | | | | | |
| rs11418337 | 14 | 37382318 | C | 0.595 | 0.980 | 1.50E-09 | 0.067 | 0.011 | *SLC25A21 (PAX9, SLC25A21, RP11-964E11.2)* |
| BMI-adjusted-habitual snoring | | | | | | | | | |
| rs8023248 | 14 | 37402131 | C | 0.569 | 0.979 | 6.90E-12 | 0.072 | 0.011 | *SLC25A21 (PAX9, SLC25A21, RP11-964E11.2)* |
| rs10788141 | 10 | 122924854 | G | 0.397 | 0.985 | 1.60E-09 | -0.064 | 0.011 | *WDR11:FGFR2* |
| 10 study areas: Habitual snoring | | | | | | | | | |
| rs11418337[a] | 14 | 37382318 | C | 0.405 | 0.981 | 2.08E-08 | 0.068 | 0.012 | *SLC25A21 (PAX9, SLC25A21, RP11-964E11.2)* |

A1: effect allele, A1FREQ: effect allele frequency in CKB, Info: imputation quality score, β: effect of effect allele on the trait, SE: standard error of the effect, eQTL: expression quantitative trait loci. Novel loci were defined as the genomic risk loci that were more than 500 kb away from the loci identified in previous GWAS for snoring and by low linkage disequilibrium $r^2 < 0.1$ between the genomic risk loci and the previous loci. rs10886864 and rs10788141 were mapped in the intergenic region of *WDR11* and *FGFR2*.
[a]The novel loci were significant [$P < 5 \times 10^{-9}$], other loci were suggestively significant [$P < 5 \times 10^{-8}$].

## Mendelian randomization

SNPs previously reported to be associated with the outcomes ($P < 1 \times 10^{-5}$) were excluded (Supplementary Data 17). The *F* statistic of each SNP was larger than ten, suggesting a low possibility of weak instrumental variable (IV) bias (Supplementary Data 18). The intercept of MR Egger regression indicated no significant horizontal pleiotropy ($P > 0.05$). Several inverse variance weighted (IVW) Cochrane's *Q* tests showed the existence of heterogeneity ($P < 0.05$). Thus, the random-effect model in IVW was applied. All analyses passed the MR-Steiger test ($P < 0.001$) (Supplementary Data 19).

Bi-directional MR for causal associations between snoring traits and the genetically correlated traits suggested that higher BMI, body weight, SBP, and DBP were causal for the increasing risks of snoring traits, the corresponding ORs (95%CIs) with IVW were 1.412 (1.277, 1.562), 1.411 (1.279, 1.557), 1.259 (1.091, 1.452), 1.347 (1.145, 1.584) (per standard deviation [SD] increased in exposures). The results were stable across sensitivity analyses. On the reverse, higher risks of snoring traits were only associated with DBP (snoring: $\beta$[95%CI] = 0.021 [0.002, 0.041], $P = 0.033$; habitual snoring: $\beta$[95%CI] = 0.012 [0.000, 0.025], $P = 0.046$, per 0.5-fold increase in the probability of the exposure), while the causal effect attenuated to null with a $P$-value $< 1 \times 10^{-5}$ as the threshold for IV selection. No other causal links were identified in the present MR analysis (Fig. 5, Supplementary Data 20).

## Discussion

To our knowledge, our study was the first GWAS of snoring in the Asian population. Four novel loci were identified, most replicated in the UKB. Snoring genes overlapped with obesity gene sets. Genetic correlations were found between snoring and general obesity, blood pressure, and smoking. Higher BMI, weight, SBP, and DBP levels were causal for snoring, and a reverse effect was observed between snoring and DBP.

We highlighted the novel genes *SLC25A21* and *PAX9* on 14q13.3, also identified in the GWAS after adjusting for the BMI, supporting the notion that the genetic architecture of snoring was partly not explained by obesity.

*SLC25A21* encoded the mitochondrial oxo-dicarboxylate carrier, which transported the precursor for acetyl-CoA. The mutation of *SLC25A21* was associated with the spinal muscular atrophy-like disease[8], which probably contributed to the decreasing forced vital capacity and dysfunction of the genioglossus muscle, related to the development of snoring. *PAX9* was a member of the paired box family of transcription factors, responsible for oligodontia[9] and nonsyndromic cleft lip[10] among the Asians in the previous studies. Our results indicated that the structural abnormality of the upper airway and craniofacial region was related to snoring among Chinese adults. This was in line with a previous finding that Chinese exhibited more craniofacial bony restriction for the etiology of snoring compared with Caucasians[11].

Novel loci rs10886864 mapped on the intergenic region between *WRD11* and *FGFR* was related to BMI[12] and lipid level[13] in the Asians, and related to type 2 diabetes in a trans-ancestry Meta-analysis[13]. The novel genes *NAA25* and *ALDH2* were related to aspartate aminotransferase[13] and aldehyde dehydrogenase 2 levels in Asians[14]. Besides, the novel gene *VTI1A* contributed to the vesicle-mediated transport and golgi-to-endoplasmic reticulum retrograde transport[15]. The results above were consistent with the fact that obesity and diabetes mellitus were related to sleep disorder breathing[2], which indicated that the dysfunction of the transportation system might contribute to the development of snoring. However, more evidence is necessary to investigate this inference.

The present study also replicated several loci identified in the UKB GWAS. *FTO* was located on the *FTO*-obesity-variant mechanism. The wild-type *FTO* gene contributed to repressing the transcription of *IRX3*, leading to mitochondrial thermogenesis and a browning adipocyte program[16]. The deposited fat surrounding the upper airway led to pharyngeal collapse[17]. *MSRB3* was related to the metabolism pathways, hippocampal volume, and cognitive dysfunction[7], *BDNF* contributed to the development of the neural system, indicating a possible role of neurological abnormalities in snoring[18].

Among East Asians, general obesity was genetically correlated with snoring, while central obesity was not. The result was inconsistent with that

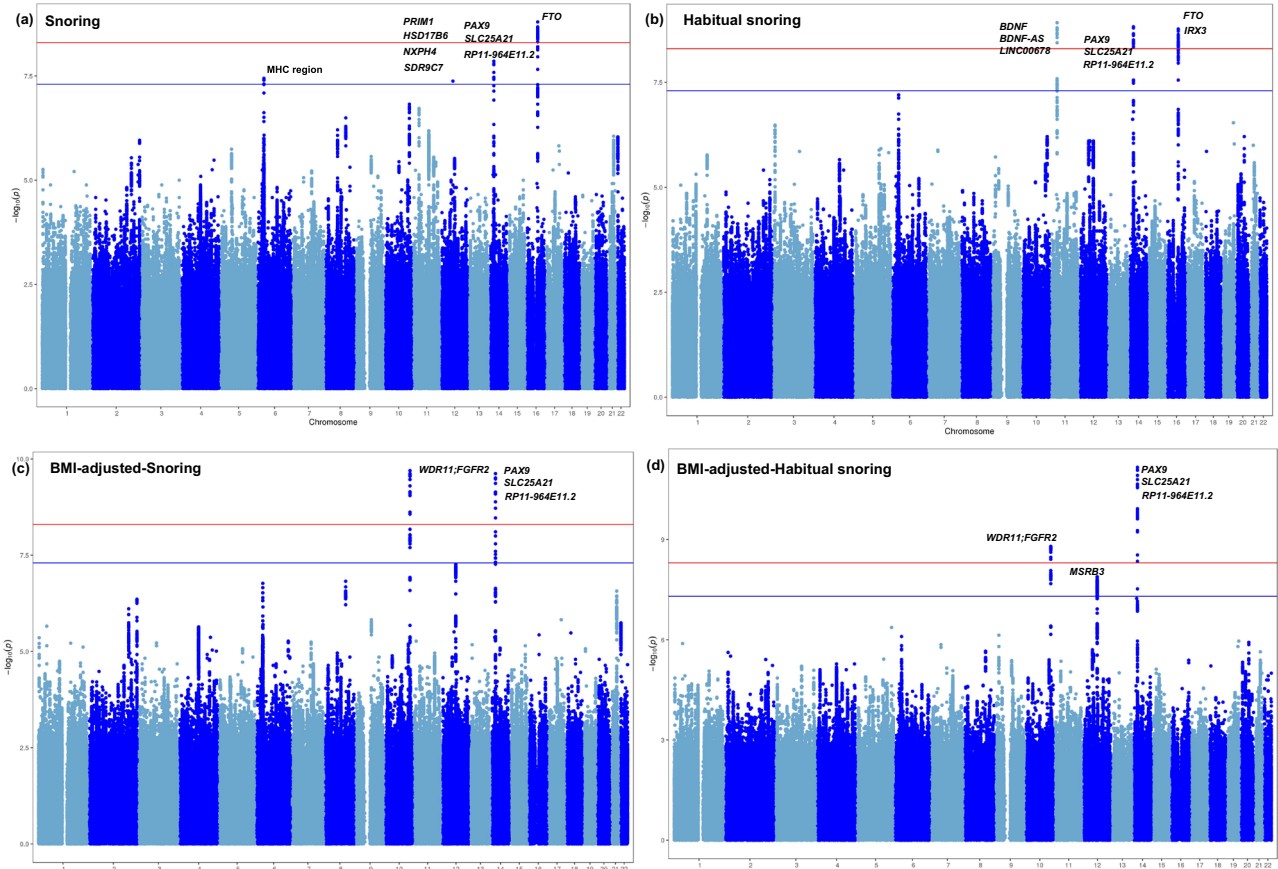

**Fig. 1 | Manhattan plots for GWAS of snoring ($N = 100,626$) and habitual snoring ($N = 76,403$) in CKB.** Manhattan plots for GWAS of snoring and habitual snoring, and the BMI-adjusted results, which identified four loci for snoring (**a**), three loci for habitual snoring (**b**), two loci for BMI-adjusted snoring (**c**), and three loci for BMI-adjusted habitual snoring (**d**). The X-axis denotes the genomic position, and the Y-axis denotes the $\log_{10}$ (*P*-value) of the association test. The genome-wide significance level ($P = 5 \times 10^{-9}$) and the suggestive significance level ($P = 5 \times 10^{-8}$) were represented by the red and blue lines, respectively. Symbols of genes mapped to the loci were marked on the plots.

previously reported in UKB[7]. However, they ignored that WC and waist-to-hip ratio (WHR) were affected by BMI[19]. Thus, WC and WHR adjusted for BMI were applied as the proxies of central obesity in our study. The present analysis conducted among the Europeans showed a positive genetic correlation between snoring, BMI, and WHRadjBMI, not WCadjBMI, which indicated the shared genetic bases of central adiposity with snoring between the two ancestries were probably different. Blood pressure levels and smoking behavior shared the genetic structure with snoring, both in the East Asian and European populations. While only the European population showed genetic correlations between the levels of glucose and lipid metabolites with snoring. Therefore, there are large differences in the shared genetic components between metabolic traits and snoring among Europeans and Asians, which needs more genetic and biological mechanism studies to confirm.

The bi-directional MR indicated that general obesity was a driving component of snoring, not a symptom or comorbidity. The findings were supported by the previous cohort study[20] and our previous work using autoregressive cross-lagged panel analysis[21]. BMI and body weight represented the whole-body fat, which contained the fat deposited surrounding the upper airway[2]. The results differed from UKB, which might indicate racial differences. In line with the previous MR study, a mutually causal relationship was observed between DBP and snoring[7]. In addition, higher SBP was also causal for snoring. The underlying mechanism of snoring-induced hypertension was mainly explained by sympathetic activation and oxidative stress due to apneic episodes[22]. Besides, fluid retention and shift to the neck at night due to hypertension was a possible mechanism on the reverse[2]. These findings suggested that maintaining the cardiometabolic factors, such as BMI and blood pressure, were beneficial for preventing snoring, and snoring could be an indicator for managing blood pressure.

Here, we compared the prediction performances of PRS on the baseline and resurvey snoring traits. Considering the ancestry differences, we focused on the PRS based on the CKB sample. The PRS of habitual snoring performed better on the resurvey trait. Snoring traits changed from the baseline survey to the second resurvey in CKB, thus, a better prediction could be observed when the same trait was applied to the base and target sample. While the PRS of snoring showed similar predictive performances between the baseline and resurvey snoring. In the present study, the heritability of habitual snoring was higher than that of snoring. Thus, the difference in prediction performance could be more pronounced for the PRS of habitual snoring than that of snoring.

Our study provided the genetic etiology of snoring and studied the causal associations between snoring and cardiometabolic traits in Asians. However, several limitations should be acknowledged. First, the sample size of GWAS was about 1/4 of UKB's, leading to fewer identified loci in CKB. Second, the loci of GWAS at the study-area level were less than the primary analysis, indicating a possible bias caused by population stratification. Nevertheless, the intercepts of LDSC were close to zero, revealing a small magnitude of bias[23]. Third, we regarded the participants who did not know their snoring status as non-snorers, which probability led to the information bias. Last, the external validity of PRS needs to be confirmed in other populations, especially for the novel loci among an independent Asian population with similar allele frequency with CKB.

The present study performed the GWAS of snoring based on 100,626 Chinese adults. Four novel loci revealed structural abnormality of the upper

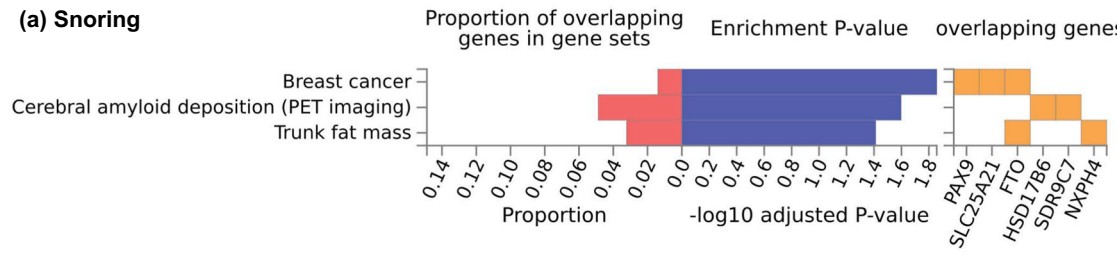

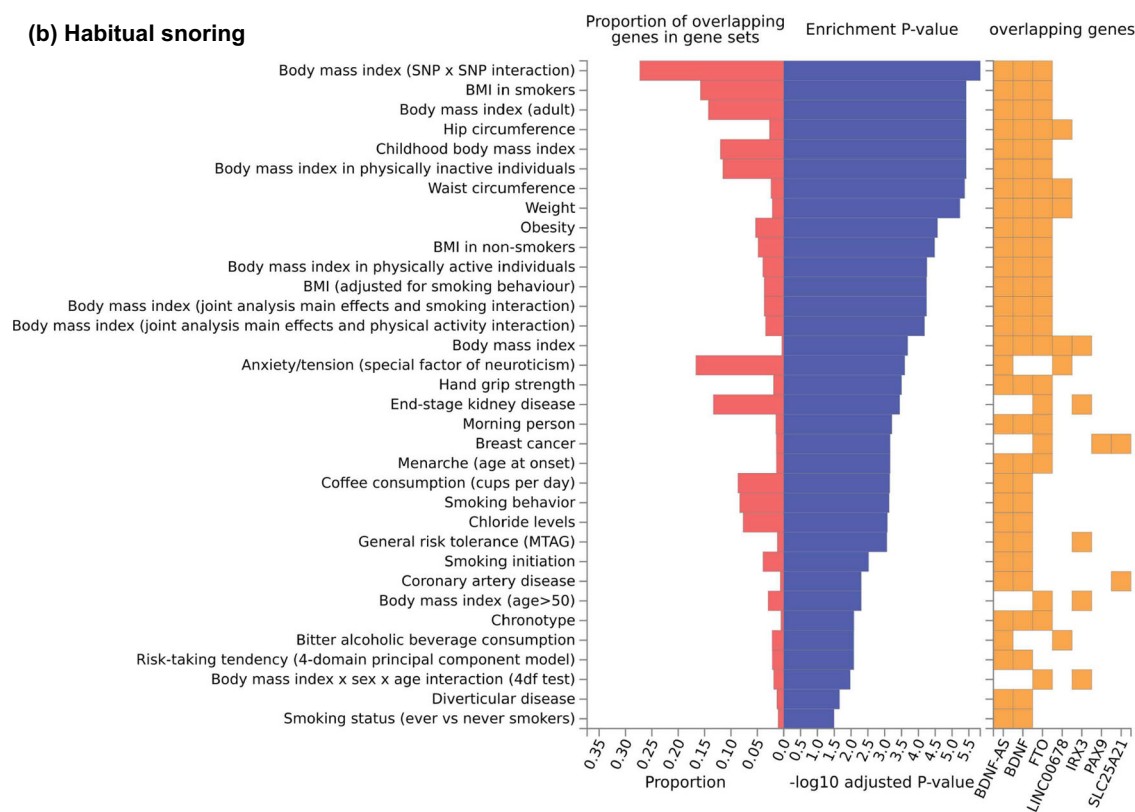

**Fig. 2 | Gene-set enrichment analysis of snoring (a) and habitual snoring genes (b).** Gene sets were obtained from the GWAS catalog. The complete results of the enrichment analysis are shown in Supplementary Data 9.

airway and craniofacial region, and dysfunction of metabolic and transport systems were the possible etiologies of snoring among Chinese adults. General obesity and higher blood pressure levels were causal for snoring, which was reversely associated with higher DBP. Our findings indicated that maintaining cardiometabolic health was essential for preventing and treating snoring, and hypertension should be considered among snorers.

## Methods
### Study design and participants
The overall study design was shown in Fig. 6. The first stage identified the genomic risk loci of snoring and habitual snoring by GWAS in the CKB population. Bi-directional replication with UKB and the PRSs of the snoring traits were conducted. The second stage was to perform the genetic correlation and bi-directional MR to estimate the genetic relationship between snoring and cardiometabolic traits with applying GWAS summary statistics from the Biobank of Japan (BBJ).

CKB study[24,25] is a prospective cohort study that recruited 512,715 adults aged 30-79 years living in 10 study areas across China (Jiangsu, Zhejiang, Sichuan, Hunan, Guangxi, Hainan, Heilongjiang, Shandong,

Gansu, and Henan, the first six areas were in southern China, others were in northern China). Extensive questionnaire data, physical measurements, and blood samples were collected upon baseline assessment in 2004–2008, led by trained investigators. Blood samples were used for genotyping. Two resurveys were conducted in 2008 and 2013-2014, which involved ~5% randomly chosen surviving participants.

The present study included participants with no missing values of snoring phenotype or genotype and passed pre-imputation QC ($n = 100,640$). Participants who failed sex QC or missed data were excluded, leaving 100,626 participants for GWAS of snoring.

The CKB study was approved by the Ethics Review Committee of the Chinese Center for Disease Control and Prevention (Beijing, China: 005/2004) and the Oxford Tropical Research Ethics Committee, University of Oxford (Cambridge, UK: 025–04). The UKB study was approved by the North West Multi-center Research Ethics Committee. The BBJ study was approved by the research ethics committees at the Institute of Medical Science, the University of Tokyo, the RIKEN Yokohama Institute, and the 12 cooperating hospitals. All participants provided written consent to participate in the study.

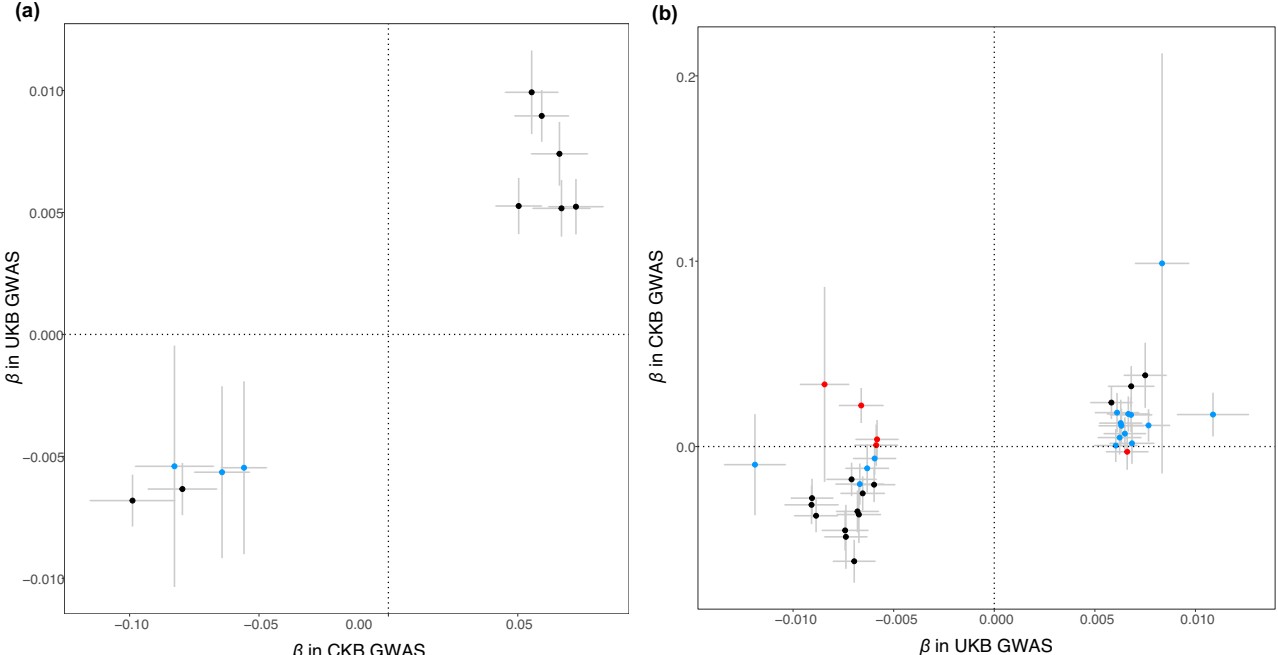

**Fig. 3 | Bidirectional replication analysis of CKB and UKB GWAS of snoring.**
CKB, China Kadoorie Biobank; UKB, UK Biobank; GWAS, genome-wide association study. Scatter plots of replication analysis for the 11 identified in CKB GWAS of snoring traits and the sensitivity analysis (**a**) and 35 loci identified in UKB GWAS of snoring (**b**). The X-axis denotes the genetic effects ($\beta$) of each SNP on the snoring traits in the discovery summary statistics, the Y-axis denotes the genetic effects ($\beta$) in the replication summary statistics, and error bars in grey denote the standard error of $\beta$. Plots in black meant the SNPs passed the replication, plots in blue meant the SNPs had no significant effect, and plots in red meant the SNPs had a reverse direction in the replication. The complete results of the replication analysis were shown in Supplementary Data 11,12.

| Trait | Snoring | P.Snoring | Habitual snoring | P.Habitual |
|---|---|---|---|---|
| BMI | | <0.001 | | <0.001 |
| Body weight | | <0.001 | | <0.001 |
| SBP | | 0.008 | | 0.034 |
| Ever smoked | | 0.025 | | 0.005 |
| DBP | | 0.109 | | 0.049 |
| HbA1c | | 0.171 | | 0.322 |
| Ever drunk | | 0.228 | | 0.228 |
| WHRadjBMI | | 0.395 | | 0.494 |
| WCadjBMI | | 0.709 | | 0.395 |
| TG | | 0.776 | | 0.392 |
| Glucose | | 0.910 | | 0.883 |
| Height | | 0.793 | | 0.694 |
| TC | | 0.171 | | 0.171 |
| LDLC | | 0.228 | | 0.494 |
| HDLC | | 0.110 | | 0.088 |

**Fig. 4 | Genetic correlations between snoring and cardiometabolic traits.** LD score-based estimates of the genetic correlation between snoring and cardiometabolic traits. BMI body mass index, WCadjBMI waist circumference adjusted for BMI, WHRadjBMI waist-to-hip ratio adjusted for BMI, HbA1c glycosylated Hemoglobin, HDLC high-density lipoprotein cholesterol, LDLC low-density lipoprotein cholesterol, TC total cholesterol, TG triglycerides, DBP diastolic blood pressure, SBP systolic blood pressure. GWAS summary statistics for the cardiometabolic traits were derived from the Biobank of Japan, except for BMI, WCadjBMI, and WHRadjBMI summary statistics from the China Kadoorie Biobank. The error bars meant the 95% confidence intervals of the corresponding $r_g$ values. The complete results of the genetic correlation analysis are shown in Supplementary Data 15.

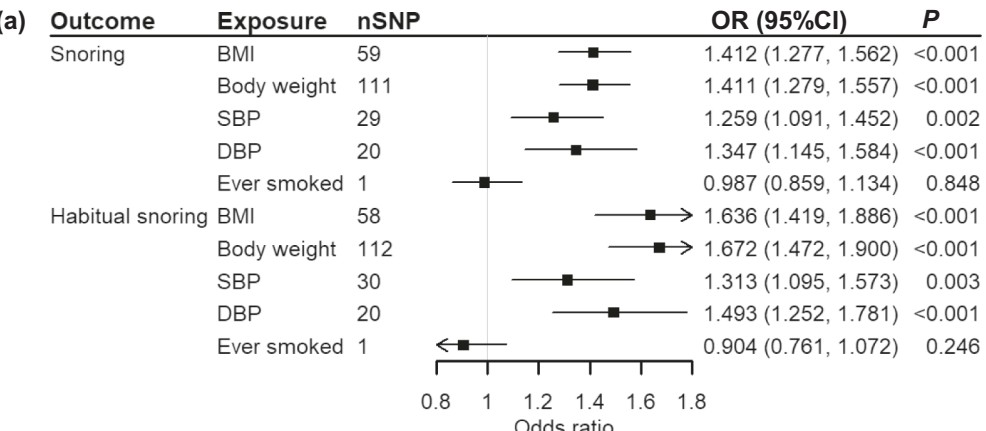

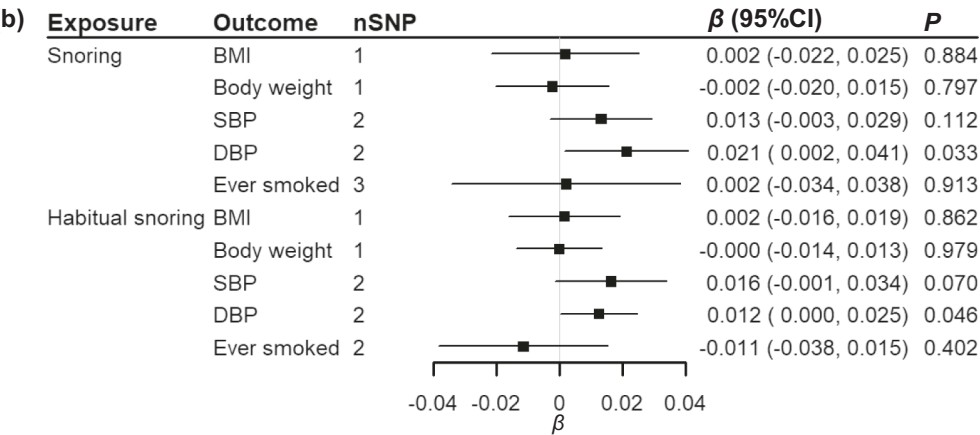

**Fig. 5 | Results of two-sample bi-directional MR.** nSNP was the number of SNPs used as the instrumental variables. BMI body mass index, SBP Systolic blood pressure, DBP Diastolic blood pressure. Causal effects of cardiometabolic traits on snoring traits were shown in panel a, in which the odds ratio was scaled to represent the association of per SD increase in the cardiometabolic index and the probability of snoring traits. The effects of snoring on cardiometabolic traits were shown in panel b, in which the beta was scaled to represent the association of a 0.5-fold increase in the prevalence of snoring or habitual snoring and the increase in the cardiometabolic index. The error bars meant the 95% confidence intervals of the corresponding odds ratios. Results with the IVW method are shown in the figure. The complete results of the MR analysis are shown in Supplementary Data 20.

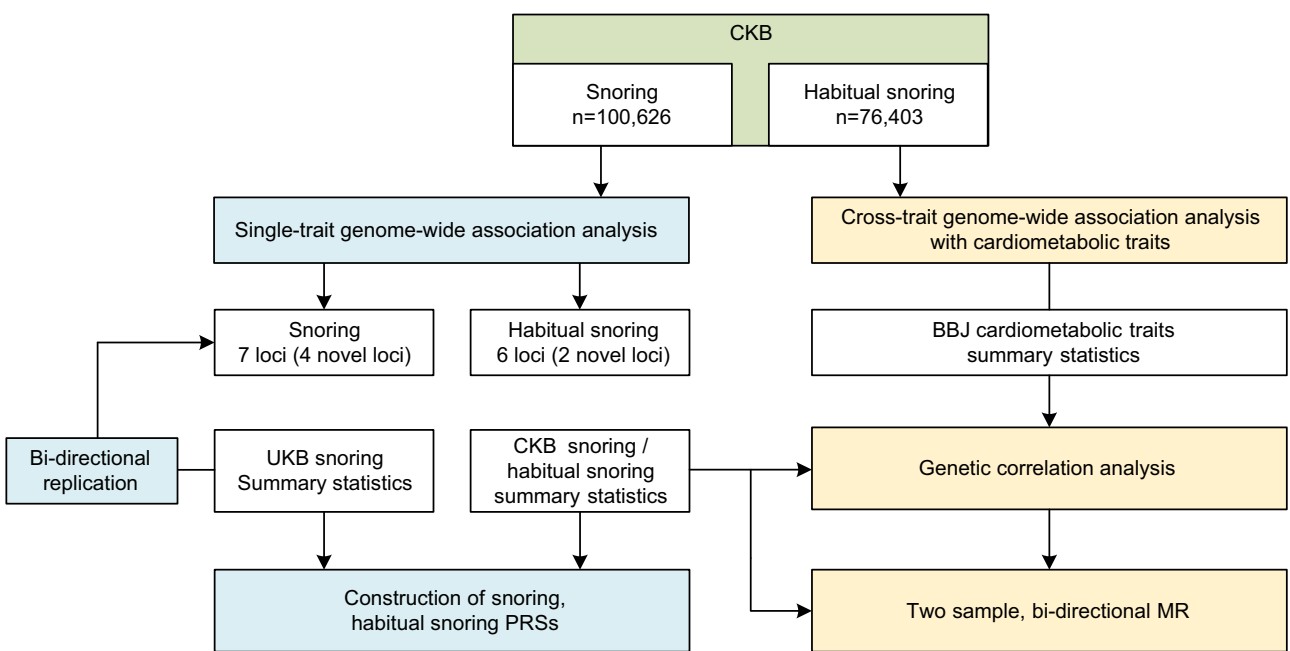

**Fig. 6 | Overall study design.** CKB China Kadoorie Biobank, UKB UK Biobank, BBJ Biobank of Japan, MR Mendelian randomization, PRS polygenic risk scores.

## Snoring

Participants self-reported the snoring trait. They were asked about their snoring habits: "Do you snore during sleep?" Three options were available: "Yes, frequently", "Yes, sometimes", or "No/Don't know." Participants with the first two options were classified into the snoring group. Furthermore, those who reported "frequently" snoring were defined as the habitual snoring group.

## Genotyping, imputation, and quality control

The CKB project used the Affymetrix Axiom array designed for Chinese Han ancestry, and the genotyping was performed by Beijing Genomics Institute (BGI, Shenzhen, China). Two batches of participants were genotyped with two versions of arrays. In 2015, more than 32,000 participants were tested using the first array type, which contained 700K SNPs. In 2016, more than 69,000 participants were tested using another array containing 803 K SNPs. Among the two batches of participants, 8143 had atherosclerotic vascular disease, 5917 had hemorrhagic stroke cases, and 5203 had chronic obstructive pulmonary disease; others were the healthy controls[26–28].

QC was performed before imputation. Participants with call rate <0.98, whose heterozygosity ≥ three SD from the mean, and whose first two principal components (PCs) ≥ four SD from means were excluded. Variants with call rate > 0.98, plate effect $P > 10^{-6}$, batch effect $P > 10^{-6}$, Hardy-Weinberg equilibrium deviations $P > 10^{-6}$, MAF difference from 1000 Genome East-Asian Phase III (1000 G EAS[29]) frequencies < 0.2 were excluded, leaving 532,415 variants shared on both versions of the array.

Qualified samples and genotypes were phased using SHAPEIT version 3. The imputation was performed using IMPUTE version 4 for each 5 Mb interval, taking the 1000G EAS[29] (released in October 2014) as the reference, resulting in 24,785,240 SNPs.

In the current study, SNPs that met any of the following criteria were excluded: (i) imputation quality score (Info) ≤ 0.3 for MAF > 3%, or Info ≤ 0.6 for MAF 1–3%, or Info ≤ 0.8 for MAF 0.5–1%, or Info ≤ 0.9 for MAF 0.1–0.5%; (2) Hardy-Weinberg equilibrium deviations $P ≤ 10^{-6}$; (3) call rate ≤ 95%; (4) SNPs on the sex chromosome[30]. Finally, 7,063,876 SNPs remained for GWAS analysis (Supplementary Fig. 5).

## Descriptive analysis

The present study described the baseline characteristics of the CKB participants included in the present GWAS analysis by their snoring statuses (non-snorers, occasional snorers, habitual snorers). Among the characteristics, the geographical origins meant the study areas collected in the baseline survey. Genetic outliers within each geographical origin were defined as those with the means of the PCs 1-11 more than three standard deviations from the region means of the PCs 1-11[31]. The linear trend was tested by treating the snoring statuses as a continuous variable.

## Genome-wide association analysis

The GWAS analysis compared the snoring group ($n = 47,208$) and the non-snoring group ($n = 53,418$) using the BOLT-LMM 2.3.2 linear mixed model, which accounted for cryptic relatedness and population stratification[32]. Associations were analyzed in additive genetic models adjusting for age, age[2], sex, study areas, genetic array types, the PCs 1–10 of ancestry at the national level, and four baseline disease statuses. We converted SNP effect size estimates ($\beta$) on the quantitative scale to traditional $\beta$ when analyzing case-control traits[33]. A similar GWAS was conducted among habitual snoring ($n = 22,985$), with the non-snoring group ($n = 53,418$) as the reference group.

Weight, height, hip circumference, and WC were measured by trained staff using well-calibrated instruments[19,25] in the baseline survey of CKB. The present study included three obesity traits (BMI, BMI-adjusted WHR [WHRadjBMI], and BMI-adjusted WC [WCadjBMI])[19,34,35]. Residuals of the traits were adjusted for age, age[2], sex, and study areas in a linear regression model, and were inverse normal transformed. GWASs of the three obesity traits were also conducted using the BOLT-LMM 2.3.2 linear mixed model, adjusting for genetic array types, PCs 1-10, and baseline disease statuses.

## Sensitivity analysis

(1) Given the strong correlation between snoring and BMI, BMI was added as a covariate in the GWAS. (2) Considering the regional differences in snoring and genetic background, we conducted the GWAS stratified by ten study areas of CKB (genetic outliers within each geographical origin were additionally excluded), adjusting the PCs at the study-area level (Supplementary Data 1). Then a fixed-effect inverse-variance-weighted Meta-analysis including ten study areas was performed using METAL[36]. (3) Considering the gender difference in snoring, GWAS analysis was conducted in males and females separately.

## Post-GWAS analysis

Genomic risk loci were identified by PLINK 1.9[37]. Considering the multiple testing burdens of different types of snoring traits and sensitivity analyses, we set the significance threshold for our GWAS at $P = 5 \times 10^{-8}/10$ ($= 5 \times 10^{-9}$) and used $P = 5 \times 10^{-8}$ as a threshold of suggestive associations. A two-step clumping method based on 1000 G EAS[29] was applied[38]. Based on GWAS summary statistics, the first step of clumping ($P$ value < $5 \times 10^{-8}$, $r^2 < 0.6$) derived independent significant SNPs. SNPs with $r^2 \geq 0.6$ with any of the detected independent significant SNPs were included for further functional annotation. Based on the independent significant SNPs, the independent lead SNPs were defined if they were independent of each other at $r^2 < 0.1$. Independent significant SNPs dependent on each other at $r^2 < 0.1$ or closer than 250 kb were assigned to the same genomic risk region. Each genomic risk region was represented by the top lead SNP with the minimum $P$-value in the region, reported as the genomic risk loci in the present study.

Novel loci were defined at two levels: locus and region. If the genomic risk locus was more than 500 kb away from the previously known loci reported in the GWAS catalog for snoring (search date: Mar 3rd, 2023), it was defined as a novel locus. To determine whether the locus was in a novel or known genomic region, we checked whether the LD $r^2 < 0.1$ between the genomic risk locus and known loci.

Both positional mapping and eQTL mapping were performed to prioritize the candidate genes. For the positional mapping, genes nearest to the genomic risk loci or any genes within a 10 kb window around the genomic risk loci were mapped. If there were no genes around the genomic risk loci, genes containing a non-synonymous SNP in high LD with the genomic risk loci ($r^2 > 0.6$ in 1000 G EAS[29]) were mapped as the candidate genes. The positional mapping and annotation were conducted via ANNOVAR[39]. The amino acid change information was obtained from RefSeq Gene and UCSC Known Gene. Besides, the VarNote platform[40] (http://www.mulinlab.org/varnote/application.html#REG) was applied as a complementary approach for the genomic risk loci, which couldn't be annotated in ANNOVAR. The eQTL mapping mapped SNPs to the protein-coding genes, which likely affected the expression of the genes up to 1 Mb (cis-eQTL). In other words, genes with cis-eQTL associated with the lead variant obtained from the Genotype-Tissue Expression project were mapped as the candidate genes. The eQTL mapping was performed using the Genotype-Tissue Expression database (http://www.gtexportal.org/home/), which contained the expression levels of 20,260 protein-coding genes across 49 tissues[41]. The major histocompatibility complex regions (the region between *MOG* and *COL11A2* genes in chromosome 6) were excluded by default and were excluded in the following analysis. Considering the multiple tests of the protein-coding genes, significant eQTLs are defined as FDR (gene $q$-value) ≤ 0.05.

The present study tested the relationship between each of the prioritized genes and tissues using FUMA GENE2FUNC[38]. The gene expression heatmap showed the average of normalized expression per tissue, which allowed the comparison of gene expression across tissues within a gene. In addition to the single gene level, we performed the gene-set enrichment analysis to explore whether the prioritized genes overlapped the gene sets from MsigDB. The latter were within the categories of the Gene Ontology, GWAS catalog, and Wiki-Pathways. FDR was used to correct for multiple tests[38].

## Bidirectional replication

The present study leveraged summary statistics of CKB and UKB GWAS of snoring to perform a bidirectional replication.

UKB habitual snoring GWAS study included participants of European ancestry. Baseline data were collected between 2006–2010. Snoring was a self-reported trait (Field-ID: 1210): "Does your partner or a close relative or friend complain about your snoring?" After excluding participants who answered "Don't know" or "Prefer not to answer", 408,317 participants were included in the GWAS, containing 37% snorers. GWAS of snoring in UKB was performed using BOLT-LMM, adjusting for age, sex, genotyping array, and the first 20 PCs as fixed effects. A post-GWAS strict QC was carried out, corresponding to minor allele frequency ≥0.005 and imputation quality score ≥0.6. Besides, similar GWASs stratified by sex, and additionally adjusted for BMI were also carried out among UKB participants[7]. The snoring GWAS summary statistics from UKB were obtained from the NHGRI-EBI Catalog (https://www.ebi.ac.uk/gwas/home)[42].

To test the validity of our results, we did an independent replication in the UKB GWAS of snoring[7]. All the genomic risk loci for snoring traits identified in the present main and sensitivity analyses were included. We confirmed that a genomic risk locus (locus) in CKB passed the replication with the following criteria: (i) the locus existed in UKB GWAS summary statistics; if not, the proxy SNP (LD $r^2 > 0.8$) should exist, (ii) the direction of $\beta$ and effect allele of the locus was matched across CKB and UKB. (iii) Considering the ancestry difference, we applied the $P$-value $< 5.00 \times 10^{-5}$ in the GWAS of UKB to determine statistical significance for the replication.

Besides, the present study performed a replication analysis for genomic risk loci identified in UKB GWAS of snoring in the summary statistics CKB GWAS of snoring. SNPs that passed the QC of CKB GWAS were included in the replication analysis. The criteria for replication were the same as above, except for the third one: as the discovery sample size of UKB was larger than that of CKB, a loose $P$-value threshold ($P < 0.05$) in GWAS of CKB was applied for the reverse replication.

Considering the ancestral difference, minor allele frequencies for the snoring loci included in the replication analysis were compared between the East Asian population from the CKB study and the European population from the UKB study[7]. A two-sided Mann-Whitney U test was applied for the statistical comparison[43].

## PRS construction

Before computing the PRS, base and target data were prepared following the tutorial of PRS analyses[44]. CKB base data (snoring: $n = 78,069$, habitual snoring: $n = 62,885$) was applied to generate the summary statistics of the two snoring traits, and the summary statistics of UKB snoring GWAS was also used[7]. The heritability of each summary statistic was more than 5%, and GWAS QC was already done with the same criteria as the present CKB main analyses and UKB study. The duplicate, ambiguous, or mismatching SNPs were excluded, leaving 5,276,463, 5,275,387, 4,373,495 variants in summary statistics of CKB snoring, CKB habitual snoring, UKB snoring for PRSs construction, respectively.

The target individuals from CKB were independent of the base dataset. For the QC of target data, the present study removed those SNPs with MAF < 0.001, Hardy-Weinberg equilibrium deviations $P < 10^{-6}$, and the mismatching or duplicate SNPs. Individuals with extreme heterozygosity (≥three SD) and individuals closely related in the sample ($\pi > 0.125$) were excluded, leaving 17,951 and 11,494 individuals in the independent target sample of snoring and habitual snoring.

PRSice-2 was used to construct and select the PRSs with the best prediction on snoring or habitual snoring at baseline or the second resurvey. A framework of PRSs construction was shown in Supplementary Fig. 6. PRSice-2 implemented the standard "clumping + threshold" method with a sequence of PLINK1.9 function and QC steps[45]. We used the clumping algorithm in PRSice-2 to clump together SNPs within 250 kb in LD with an $r^2 > 0.1$. Based on a range of $P$-value thresholds for SNP ($5 \times 10^{-8}$ - 1, interval:0.0001), SNPs were extracted from the summary statistics (snoring in CKB: 1-261,462 SNPs, habitual snoring in CKB: 0-260,718 SNPs, snoring

in UKB: 59-131,166 SNPs), then the best model was derived according to Nagelkerke's $R^2$ value, and the PRS construction models included the following covariates: age, age[2], sex, study areas, PCs 1-10, genotyping array, and baseline disease status. Last, $R^2$ values between the best models of the CKB and UKB of the same traits were compared, and the larger $R^2$ was chosen for the final model corresponding to each snoring trait.

## Genetic correlation analysis

LDSC (version 1.0.1)[23,46] was applied to estimate SNP-based heritability and genetic correlation by using LD scores calculated from the 1000G EAS[29]. GWAS summary statistics from CKB and BBJ[47,48] were applied to investigate the genetic correlations between the two snoring traits and 13 cardiometabolic traits among East Asians (Supplementary Data 21).

The BBJ study[47,48] was conducted among the East Asian population, with sample QC of age ≥18, weight and height data registered and within threefold the interquartile range, call rate>0.98, not closely related sample, not outliers in PC analysis. The imputation was performed with the whole genome sequencing data from the BBJ ($N = 1037$) and the East Asian sample of 1000 Genomes Project Phase I v3 reference panel. QC for variants in BBJ used the following criteria: sample call rate < 0.98, SNP call rate < 0.99, Hardy-Weinberg equilibrium deviations $P < 1 \times 10^{-6}$, number of heterozygotes < 5 and imputation quality score < 0.7 were excluded. All the GWAS association analyses were performed using BOLT-LMM[32], except for the GWAS of BMI performed with mach2qtl[12], and the GWAS of ever-smoked performed with the SAIGE[49]. The sample sizes of the GWASs and adjustments in the GWAS models were shown in Supplementary Data 21. The GWAS summary statistics from BBJ were available from BBJ PheWeb (https://pheweb.jp/)[12,13,49,50]. FDR was used to correct for multiple tests. The cardiometabolic traits genetically correlated with snoring traits were further estimated for the causal associations.

Besides, genetic correlation analysis between snoring and cardiometabolic traits was conducted within the European population from the UKB study, based on the GWAS summary statistics obtained from Neale Lab (http://www.nealelab.is/uk-biobank) (GWAS round 2) and NHGRI-EBI Catalog (https://www.ebi.ac.uk/gwas/home) (Supplementary Data 22)[42].

## Mendelian randomization

A bi-directional MR was conducted to estimate the causal relationship between cardiometabolic and snoring traits. GWASs of two independent samples of East Asia, BBJ[12,13,47–49] and CKB, were leveraged to avoid overlapping subjects or the possibility of population stratification.

Based on the GWAS summary statistic of snoring and cardiometabolic traits, the present study selected the eligible IVs following a series of QC steps. First, SNPs associated with the traits at a genome-wide significant level ($P < 5 \times 10^{-8}$) were selected, and those associated with snoring and ever smoked at $P < 5 \times 10^{-8}$ and $P < 1 \times 10^{-5}$ were selected[51]. Second, clumping ($r^2 < 0.001$, window = 10,000 kb[52]) was performed with the 1000 G EAS[29]. Only SNPs with the lower P could remain among the SNPs in LD. Third, SNPs with MAF < 0.001 were excluded. Fourth, SNPs in the major histocompatibility complex regions were removed. Fifth, SNPs should be extracted from the outcome GWAS summary statistic (harmonization). If the SNPs didn't exist in the outcome data, the effect of proxy SNP in strong LD (via PLINK clumping function: $r^2 < 0.8$, window = 1000, based on 1000 G EAS[29]) was used. For the strand issues, the ambiguous SNPs (with non-concordant alleles) and palindromic-not inferable SNPs were excluded. Also, MR pleiotropy residual sum and outlier (MR-PRESSO) test[53] were performed, and the outlier SNPs were excluded ($P < 0.05$) to eliminate the effects of pleiotropy. Last, the reported traits of each SNP were looked up in the PhenoScanner[54,55], and those with traits related to the outcome in MR analysis were excluded. Besides, the $F$ statistic of each SNP was calculated by ($\beta$/standard error)[2] to avoid the weak IV bias[56,57].

For the MR analysis, the random-effect IVW approach[58] was used in the primary analysis, weighted median estimation[59], and MR Egger[60] as sensitivity analysis. Considering the IVs strongly associated with the

exposure might independently influence levels of a causal risk factor of the outcome, known as horizontal pleiotropy. IVW could show an unbiased result if there was balance or no horizontal pleiotropy[61]. The weighted median estimation provided an unbiased result when no more than 50% of invalid IVs (i.e., due to pleiotropy)[59]. The intercept of MR Egger regression could indicate the degree of directional horizontal pleiotropy[60]. Besides, MR-Steiger tests the causal direction[62]. Cochrane's $Q$ test assessed the heterogeneity among IVs, and a random-effect model in IVW was applied if heterogeneity existed ($P < 0.05$).

R packages "TwoSampleMR (version 0.5.6)"[61] and "MRPRESSO (version 1.0)[53]" were used for MR analysis.

## Statistics and reproducibility

The descriptive analysis was conducted with STATA version 16.0. The GWAS analysis used the BOLT-LMM 2.3.2 linear mixed model[32,33]. Fixed-effect inverse-variance-weighted Meta-analysis including ten study areas was conducted using METAL[36]. Genomic risk loci were identified by PLINK 1.9[37]. The positional mapping and annotation were conducted via ANNOVAR[39]. The eQTL mapping was conducted using the Genotype-Tissue Expression database (http://www.gtexportal.org/home/). The relationship between each of the prioritized genes and tissues, and enrichment analysis were conducted with FUMA GENE2FUNC[38]. A two-sided Mann-Whitney U test for MAF comparison was conducted with R 4.0.5[43]. PRSice-2 was used for PRS conduction. LDSC (version 1.0.1)[23,46] was applied to estimate SNP-based heritability and genetic correlation. R packages "TwoSampleMR (version 0.5.6)"[61] and "MRPRESSO (version 1.0)[53]" were used for MR analysis. The specific details of the analyses and sample sizes are described in the methods above.

## Reporting summary

Further information on research design is available in the Nature Portfolio Reporting Summary linked to this article.

## Data availability

The GWAS summary statistics from China Kadoorie Biobank (CKB) in the present study have been deposited in the Genome Variation Map (GVM)[63] in National Genomics Data Center, Beijing Institute of Genomics, Chinese Academy of Sciences and China National Center for Bioinformation[64], under the project number PRJCA023790 and accession number GVP000023. The GWAS summary statistics are publicly available in https://bigd.big.ac.cn/gvm/getProjectDetail?Project=GVP000023. The individual-level data of CKB are controlled-access and are available via an application on request. The access policy and procedures of the CKB data are available at www.ckbiobank.org. GWAS summary statistics from Biobank of Japan (BBJ) were available from BBJ PheWeb (https://pheweb.jp/), and the corresponding phenotypes and PMID of the published studies were shown in the Supplementary Data 21. GWAS summary statistics from the UK Biobank were available from the publicly available NHGRI-EBI Catalog (https://www.ebi.ac.uk/gwas/downloads/summary-statistics) and Neale Lab (http://www.nealelab.is/uk-biobank) (GWAS round 2). The study accession IDs on the NHGRI-EBI Catalog were GCST009760, GCST90020029, GCST90020025, and the Phenotype codes on the Neale Lab website could be found in the Supplementary Data 22. Source data underlying Figs. 2, 3, 4, 5 are presented in the Supplementary Data 9, 11-12, 15, 20.

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

## Acknowledgements

The most important acknowledgment is to the participants in the study and the members of the survey teams in each of the ten study areas, as well as to the project development and management teams based at Beijing, Oxford and the ten regional centres. This work was supported by the National Natural Science Foundation of China (82388102, 82192901, 82192904, 82192900). The CKB baseline survey and the first re-survey were supported by a grant from the Kadoorie Charitable Foundation in Hong Kong. The long-term follow-up is supported by grants from the UK Wellcome Trust (212946/Z/18/Z, 202922/Z/16/Z, 104085/Z/14/Z, 088158/Z/09/Z), grants (2016YFC0900500) from the National Key R&D Program of China, National Natural Science Foundation of China (81390540, 91846303, 81941018), and Chinese Ministry of Science and Technology (2011BAI09B01). The funders had no role in the study design, data collection, data analysis and interpretation, writing of the report, or the decision to submit the article for publication. Sponsors had no role in the study design, data collection, data analysis and interpretation, writing of the report, or the decision to submit the article for publication.

## Author contributions

LL, JL conceived and designed the study. LL, ZC and JC, members of the China Kadoorie Biobank Steering Committee, designed and supervised the whole study, obtained funding, and, together with CY, DS, PP, LY, YC, HD, IM, RW, FL, RS acquired the data. YZ and ZZ analyzed the data. YZ drafted the manuscript. ZZ, DS and CY helped to interpret the results. CY contributed to the critical revision of the manuscript for important intellectual content. All authors reviewed and approved the final manuscript. CY is the guarantor.

## Competing interests

The authors declare no competing interests.

## Additional information

## the China Kadoorie Biobank Collaborative Group

**Yunqing Zhu**[1], **Zhenhuang Zhuang**[1], **Jun Lv**[1,2,3], **Dianjianyi Sun**[1,2,3], **Pei Pei**[2], **Ling Yang**[4,5], **Iona Y. Millwood**[4,5], **Robin G. Walters**[4,5], **Yiping Chen**[4,5], **Huaidong Du**[4,5], **Fang Liu**[6], **Rebecca Stevens**[5], **Junshi Chen**[7], **Zhengming Chen**[5], **Liming Li**[1,2,3] & **Canqing Yu**[1,2,3] ✉

A full list of members and their affiliations appears in the Supplementary Information.

