## [Peer Review File · Communications Biology]

Reviewers' comments:

Reviewer #1 (Remarks to the Author):

The authors presented a GWAS of snoring in a large Chinese sample. The study identified novel loci and potential etiology and risk factors for snoring. Overall, the question is interesting, and the study is well conducted. However, the writing should be made clearer and more accurate.

1. The authors should be clearer about the GWAS significance thresholds in Results, Methods, and Table 1. From Methods – Post GWAS, it seems the threshold is $p < 5e-8$. However, since the study is testing multiple traits (snoring, habitual snoring, with and without adjusting for BMI), the threshold should be made more stringent to reflect multiple comparisons at the trait level.
2. For the Post-GWAS analysis, the paper should be clearer on how SNPs were mapped to genes. Do positional mapped genes refer to the nearest genes? How was eQTL used to map variants to genes (e.g., are they eGenes of the GWAS significant SNPs, and in what tissues)?
3. Line 127: "Habitual snoring genes showed relationships with obese gene sets" is vague. The sentence should be more specific that this is gene-set enrichment analysis. "Obese" should be changed to "obesity".
4. Lines 144-154, PRS for snoring: the discrepancy between PRS at baseline and PRS at resurvey was not discussed. The authors should provide more justification on fitting PRS for the two stages separately, and discuss whether/why the prediction performance differs.
5. Line 198: It is a bit of a stretch to state that dysfunction of metabolic and transport systems might contribute to snoring simply based on shared genetic locus.
6. Change section title "Discussions" to "Discussion"

Reviewers #2-3 (Remarks to the Author):

Zhu and colleagues present a genome-wide association study of snoring and habitual snoring in the China Kadoorie Biobank (CKB) with follow-up analyses presented based on the UK Biobank (UKB) and Biobank Japan (BBJ). Overall, the study uses standard methods and has some interesting findings which are organized in a reasonably clear fashion. After carrying out a follow-up Mendelian Randomization analysis, they demonstrate a potential relationship between snoring and cardiometabolic traits. While we recognize these strengths, we identified multiple areas for improvement in the manuscript related to (a) clarity and precision in the statistical methods and results presented, (b) need for more rigorous approach in the assignment of SNPs to candidate genes, (c) more systematic rationale for the overall study design and selection of cohorts for incorporation in follow-up replication and Mendelian randomization analysis, and (d) use of English language. We provide our detailed comments below.

Major:

1. Abstract: Suggest incorporating some information regarding the specific identified genes within the text of the Abstract.
2. Results: Please describe in the main text the basic descriptive statistics for the participants and include a summary table for the descriptives. This description should include a summary of the ancestral and geographic origins of the participants.
3. Results – GWAS for snoring and habitual snoring:
 - a. Please clarify what is meant by the terms "loci" and "regions" as these terms are used very specifically in presentation of the Results, but they are not defined precisely.

- b. The text in this section seems to present a combined summary of GWAS results from the overall analysis, in addition to the signals identified in sensitivity analyses (BMI-adjusted or sex stratified). However, the text does not explain that the description combined results from both primary and sensitivity analyses. Please update the text to clarify.
- c. In counting the number of loci reported (from Table 1), it appears that in some cases, a single SNP (e.g. rs712398) was counted multiple times because it was identified in both primary and BMI adjusted analyses. This approach to counting loci incorrectly inflates the number of loci identified. The method of counting loci should be refined and clarified.
4. Table 1 – Please clarify methods regarding how GWAS signals were assigned to genes as reported in the column labeled “Positional mapped genes”. There is some text provided in the Methods, but it lacks sufficient detail.
5. Methods / Results: While the authors suggest overlap between GWAS variants and eQTL to identify candidate genes, it is unclear what approach was used to account for multiple comparisons in these analyses. Also, please clarify what eQTL resources and tissues were examined in these analyses. Additionally, the overlap should be carried out more formally using integrative analysis approach such as colocalization (e.g. R/coloc package; https://cran.r-project.org/web/packages/coloc/vignettes/a01_intro.html).
6. Methods / Results – Bidirectional replication:
- How many variants were carried forward for replication analysis and how were they selected?
 - Please clarify the rationale for the threshold used in determining statistical significance for replication. Why didn't the authors simply apply Bonferroni correction for the number of variants examined for replication.
 - Why was the replication performed in UKB but not in Biobank Japan (BBJ)?
 - As the replication is done from biobanks of different geographic and ancestral representation, we suggest comparing systematically the allele frequencies for variants of interest across the biobanks examined.
7. Methods / Results – Genetic correlation analysis: Why was this analysis done on CKB and BBJ but not UKB? It would be valuable to incorporate comparison with genetic correlation of the same traits in UKB if the appropriate data are available.
8. Methods- Snoring: Please clarify how the snoring phenotype was assessed in the discovery CKB and replication UKB participants, as well as for the BBJ examined for genetic correlation. Were there any differences in questionnaire methodology in CKB vs. UKB and BBJ, or were the questions exactly the same?
9. Methods – Genome-wide association analysis: The adjustment for ten PCs of ancestry seems arbitrary. I suggest the authors examine the PCs in the population analyzed and use a systematic method to determine the appropriate number of PCs to be adjusted, as described in this reference: <https://genomebiology.biomedcentral.com/articles/10.1186/s13059-022-02761-4>
10. Methods / Results – Mendelian Randomization: Please provide a description of what approaches were used to check the assumptions of Mendelian Randomization, and summarize evidence that the assumptions were met – especially the horizontal pleiotropy assumption.
11. Supp Table 1:
- The authors state Biobank Japan (BBJ) was used to examine genetic correlation of snoring with other cardiometabolic traits, but snoring is not included in Supp Table 1. Were the GWAS summary stats of snoring available in BBJ?
 - Methods – Sensitivity analysis refers to Supp Table 1, but it seems the authors referred to the wrong Table because Supp Table 1 is not relevant to the text in that section.
 - Supp Table 1 seems out of place since it is not the first Supp Table referenced in the manuscript.

Minor:

- Abstract: Check grammar and tense usage, as some past tense should be changed to present:
 - First sentence: “was” → “is”
 - Last sentence: “revealed” → “reveal”; “provided” → “provide”

2. Introduction: Check grammar
3. Other sections of the manuscript also have awkward use of English grammar. Suggest the authors work with a native English speaker or a professional editor to improve the quality and clarity of writing.
4. Methods – Study Design: “polygenetic risk scores” -> “polygenic risk scores”

Response Letter

Dear Reviewers,

Manuscript ID COMMSBIO-23-1935 entitled "**Genome-wide association study of snoring and its associations with cardiometabolic traits**"

The authors would like to thank the reviewers for your valuable time and helpful comments. We appreciate the inputs you have given and that your inputs will definitely help improve our manuscript. The next section contains our point-by-point responses. We included both clean and tracked changes versions of the manuscript and supplementary file. Page and line numbers indicated in this response letter are based on the clean version.

We look forward to hearing from you regarding our submission. We would be glad to respond to any further questions and comments that you may have.

Sincerely,

Canqing Yu, MD, PhD

Associate Professor,

Department of Epidemiology & Biostatistics

Peking University Health Science Center

38 Xueyuan Road, Beijing 100191, P. R. China

Fax number: 86-10-82801620

Phone: 86-10-82801528 ext.322

Email: yucanqing@pku.edu.cn

Reviewer #1

Comment #1: The authors should be clearer about the GWAS significance thresholds in Results, Methods, and Table 1. From Methods – Post GWAS, it seems the threshold is $p < 5 \times 10^{-8}$. However, since the study is testing multiple traits (snoring, habitual snoring, with and without adjusting for BMI), the threshold should be made more stringent to reflect multiple comparisons at the trait level.

Response:

We agree the reviewer with the comment on p value threshold. Considering the multiple testing on different types of snoring traits and sensitivity analyses, we set the significance threshold for our GWAS at $P = 5 \times 10^{-8} / 10 (= 5 \times 10^{-9})$, and used $P = 5 \times 10^{-8}$ as a threshold of suggestive associations. As a result, among the eight loci across seven regions for the snoring traits, three loci across three regions were significant, and others were suggestively significant. We amended this in the “Methods” (Page 17, Line 16-19) and “Result” (Page 6, Line 11-13) sections.

Comment #2: For the Post-GWAS analysis, the paper should be clearer on how SNPs were mapped to genes. Do positional mapped genes refer to the nearest genes? How was eQTL used to map variants to genes (e.g., are they eGenes of the GWAS significant SNPs, and in what tissues).

Response:

For the positional mapping, genes nearest to the genomic risk loci were within a 10kb window. If there were no genes around the genomic risk loci, the genes containing a non-synonymous SNP in high linkage disequilibrium (LD) with the genomic risk loci ($r^2 > 0.6$ in 1000G EAS) were mapped as the candidate genes. The eQTL mapping mapped SNPs to the genes, which likely affected the expression of the genes up to 1Mb (cis-eQTL). In other words, genes with cis-eQTL associated with the lead variant obtained from the Genotype-Tissue Expression (GTEx) project were mapped as the candidate genes.

We have clarified the definition of positional mapping and eQTL mapping in the “Methods-Post-GWAS analysis” (Page 18, Line 4-20).

Comment #3: Line 127: “Habitual snoring genes showed relationships with obese gene sets” is vague. The sentence should be more specific that this is gene-set enrichment analysis. “Obese” should be changed to “obesity”.

Response:

We agree with the reviewer. According to the instructions of FUMA platform¹, gene sets enrichment analysis was performed to test if genes of interest are overrepresented in any of the pre-defined gene sets. Thus, we have modified the explanation of gene set enrichment analysis in the “Results” section (Page 7, Line 16-22), and we have changed the “obese” to “obesity” (Page 7, Line 20).

Comment #4: Lines 144-154, PRS for snoring: the discrepancy between PRS at baseline and PRS at resurvey was not discussed. The authors should provide more justification on fitting PRS for the two stages separately, and discuss whether/why the prediction performance differs.

Response:

We thank the reviewer for pointing this out. Considering the ancestry differences, we focused on the PRS based on the CKB sample. For the habitual snoring trait, the PRS performed better on the resurvey snoring (baseline: $R^2_{PRS}=0.0158$, resurvey: $R^2_{PRS}=0.0128$). Snoring traits changed from the baseline survey to the second resurvey in CKB (Snoring: $Kappa=0.445$). Thus, a better prediction could be observed when the same trait was applied to the base and target samples. For the snoring trait, the PRS showed similar predictive performance between the baseline and the resurvey snoring (baseline: $R^2_{PRS}=0.0066$, resurvey: $R^2_{PRS}=0.0067$). As is shown in the “Result” section (Page 6, Line 19-20), the heritability of habitual snoring (16.9%) was higher than that of snoring (10.5%). So, the effect and the difference in prediction performance between baseline and resurvey snoring could be more pronounced for the PRS of habitual snoring than that of snoring.

We addressed this in the “Discussion” section (Page 12 Line 17-22, Page 13 Line 1-3).

Comment #5: Line 198: It is a bit of a stretch to state that dysfunction of metabolic and transport systems might contribute to snoring simply based on shared genetic locus.

Response:

We agree with the reviewer that the present discussion should be more conservative. The novel genes *WRD11* and *FGFR*, *NAA25*, and *ALDH2* were related to obesity and diabetes mellitus, consistent with the previous studies that reported that these factors were related to sleep-disorder breathing². The novel gene *VTIIA* was reported to be associated with the golgi transportation³. Since none of the previous studies reported the relationship between snoring and the transportation system, more evidence is necessary to infer that the dysfunction of the transportation system might contribute to the development of snoring.

We addressed this in the “Discussion” section (Page 11, Line 9-13).

Comment #6: Change section title “Discussions” to “Discussion”

Response:

Done (Page 10, Line 10).

Reviewers #2-3 (Remarks to the Author):

Zhu and colleagues present a genome-wide association study of snoring and habitual snoring in the China Kadoorie Biobank (CKB) with follow-up analyses presented based on the UK Biobank (UKB) and Biobank Japan (BBJ). Overall, the study uses standard methods and has some interesting findings which are organized in a reasonably clear fashion. After carrying out a follow-up Mendelian Randomization analysis, they demonstrate a potential relationship between snoring and cardiometabolic traits. While we recognize these strengths, we identified multiple areas for improvement in the manuscript related to (a) clarity and precision in the statistical methods and results presented, (b) need for more rigorous approach in the assignment of SNPs to candidate genes, (c) more systematic rationale for the overall study design and selection of cohorts for incorporation in follow-up replication and Mendelian randomization analysis, and (d) use of English language. We provide our detailed comments below.

Major:

Comment #1: Abstract: Suggest incorporating some information regarding the specific identified genes within the text of the Abstract.

Response:

We added the specific genes in the “Abstract” as suggested (Page 4, Line 5-6).

Comment #2: Results: Please describe in the main text the basic descriptive statistics for the participants and include a summary table for the descriptives. This description should include a summary of the ancestral and geographic origins of the participants.

Response:

We added a table describing the baseline characteristics of CKB participants in the present study (Supplementary Table 1). Among the characteristics, geographical origins meant the study regions collected in the baseline survey. According to our recent publication⁴, participants with non-local ancestry were defined as those with the means of the first 11 principal components (PCs 1-11) more than three standard deviations from the region means of the PCs 1-11. Besides, the linear trend was tested by treating the snoring statuses as a continuous variable. We observed that the habitual snorers were more likely to be elders, males, with geographical origins in the south of China, with the same geographical and ancestry origins, with higher BMI, WC, and blood pressure, and more likely to be weekly drinkers and current smokers (all $P < 0.05$).

We addressed this in the “Methods-Descriptive analysis” (Page 16, Line 5-11) and “Results - GWAS for snoring, habitual snoring” section (Page 6, Line 3-9).

Comment #3: Results – GWAS for snoring and habitual snoring:

Please clarify what is meant by the terms “loci” and “regions” as these terms are used very specifically in presentation of the Results, but they are not defined precisely.

Response:

Yes, we agree that “loci” and “region” should be clarified. The significant independent SNPs ($P\text{-value} < 5 \times 10^{-8}$, $r^2 < 0.6$) closer than 250kb ($r^2 < 0.1$) were considered in the same

genomic region (reference panel: 1000 Genomes Phase 3 [East Asian]⁵). The genomic risk locus was the SNP with the lowest P-value within a genomic region.

We revised the description in the “Methods - Post-GWAS analysis” (Page 17 Line 14-22, Page 18 Line 1-2) and “Results - GWAS for snoring, habitual snoring” (Page 6, Line 11-19) sections. and we have modified the word “region” to “genomic region” in the manuscript.

Comment #4: The text in this section seems to present a combined summary of GWAS results from the overall analysis, in addition to the signals identified in sensitivity analyses (BMI-adjusted or sex stratified). However, the text does not explain that the description combined results from both primary and sensitivity analyses. Please update the text to clarify.

Response: We have addressed this in the “Result - GWAS for snoring, habitual snoring” section (Page 6, Line 11).

Comment #5: In counting the number of loci reported (from Table 1), it appears that in some cases, a single SNP (e.g. rs712398) was counted multiple times because it was identified in both primary and BMI adjusted analyses. This approach to counting loci incorrectly inflates the number of loci identified. The method of counting loci should be refined and clarified.

Response: We agreed that the SNPs in the same genomic region should be counted as single loci. We have modified the number of loci in the Results section (Page 6, Line 11-16), Conclusion (Page 13 Line 14-21), and Abstract (Page 4, Line 2-14) sections.

Comment #6: Table 1 – Please clarify methods regarding how GWAS signals were assigned to genes as reported in the column labeled “Positional mapped genes”. There is some text provided in the Methods, but it lacks sufficient detail.

Response: For the positional mapping, genes nearest to the genomic risk loci, or any genes within a 10kb window around the genomic risk loci were mapped. If there were no genes around the genomic risk loci, genes containing a non-synonymous SNP in

high linkage disequilibrium (LD) with the genomic risk loci ($r^2 > 0.6$ in 1000G EAS) were mapped as the candidate genes. The positional mapping and annotation were conducted via ANNOVAR⁹. The amino acid change information was obtained from RefSeq Gene and UCSC Known Gene. Besides, the VarNote platform¹⁰ (<http://www.mulinlab.org/varnote/application.html#REG>) was applied as a complementary approach for the genomic risk loci which couldn't be annotated in ANNOVAR.

We have clarified the detailed procedures for the positional mapping in the “Methods – Post-GWAS analysis” (Page 18, Line 6-15).

Comment #7: Methods / Results: While the authors suggest overlap between GWAS variants and eQTL to identify candidate genes, it is unclear what approach was used to account for multiple comparisons in these analyses. Also, please clarify what eQTL resources and tissues were examined in these analyses. Additionally, the overlap should be carried out more formally using integrative analysis approach such as colocalization (e.g. [R/coloc package; https://cran.r-project.org/web/packages/coloc/vignettes/a01_intro.html](https://cran.r-project.org/web/packages/coloc/vignettes/a01_intro.html)).

Response:

We thanked the reviewer for pointing these questions out. We thought the aims of the eQTL-mapping and colocalization analysis were different. The eQTL-mapping focused on whether the genomic risk loci could affect the expression levels of their close genes ($\pm 1\text{Mb}$) in specific tissues. If so, the loci would be mapped to the genes. A detailed methods can be found in the Functional Mapping and Annotation (FUMA) platform (<https://fuma.ctglab.nl/tutorial>) and the published paper for FUMA¹. On the other hand, the colocalization analysis tests whether the SNPS that affect snoring overlap with those that affect gene expression in specific tissues (<https://chr1swallace.github.io/coloc/index.html>).

The present eQTL analysis was based on the GTEx database (<http://www.gtexportal.org/home/>), which contained the expression levels of 20,260 protein-coding genes across 49 tissues⁶.

As to multiple testing in the eQTL analysis, the significant eQTLs were defined as false discovery rate (FDR) (gene q-value) ≤ 0.05 , accounting for the testing for 20,260 protein-coding genes across 49 tissues. As a result, fewer genes were mapped in the eQTL analysis (see the Result section).

We have addressed this in the “Methods – Post-GWAS analysis” (Page 18 Line 11-20).

Comment #8: Methods / Results – Bidirectional replication:

a. How many variants were carried forward for replication analysis and how were they selected?

Response:

We thank the reviewer’s comment on the selection criteria. For the replication analysis of snoring loci in CKB, we included 11 genomic risk loci for snoring traits identified in the present main and sensitivity analyses. Besides, the present study performed a replication analysis for genetic risk loci identified in UKB GWAS of snoring in the summary statistics CKB GWAS of snoring. A total of 35 SNPs that passed the QC of CKB GWAS were included in the replication analysis.

We have modified the Methods (Page 19 Line 12-22) and Results (Page 8 Line 1-2, 12-13).

Comment #9: b. Please clarify the rationale for the threshold used in determining statistical significance for replication. Why didn’t the authors simply apply Bonferroni correction for the number of variants examined for replication.

Response:

By now, only one GWAS of snoring was published based on the UKB population. Considering that the ancestry difference might influence the result of replication, we applied the P-value $< 5.00 \times 10^{-5}$ to determine statistical significance for the replication of the loci identified in the CKB. While the discovery sample size of UKB (n=408,317) was larger than that of CKB (n=100,626), a loose (P<0.05) was applied for the reverse replication.

Comment #10: c. Why was the replication performed in UKB but not in Biobank Japan (BBJ)?

Response:

Yes, the BBJ would be more suitable for our current replication. However, the BBJ hasn't published the GWAS nor released the GWAS summary data of snoring yet. So we performed the replication in UKB.

Comment #11: d. As the replication is done from biobanks of different geographic and ancestral representation, we suggest comparing systematically the allele frequencies for variants of interest across the biobanks examined.

Response:

We agreed that the allele frequencies should be compared systematically. We performed a trans-ancestry MAF comparison for the snoring loci in the replication analysis between CKB and UKB samples. A two-sided Mann-Whitney U test was applied for the statistical comparison⁷. The trans-ancestry MAF comparison showed that most snoring loci had higher MAF in CKB than in the UKB population ($P=0.0336$). Especially, the nonreplicated loci were likely due to a relatively low allele frequency (<0.03) among the UKB. The MAF difference between CKB and UKB was more pronounced among the SNPs identified in the UKB population ($P=0.0009$).

We added these in the “Methods- Bidirectional replication” (Page 20, Line 1-3), “Results - Bidirectional replication” (Page 8 Line 7-11, 15-17), and Supplementary Table 10, Supplementary Figure 3.

Comment #12: Methods / Results – Genetic correlation analysis: Why was this analysis done on CKB and BBJ but not UKB? It would be valuable in incorporate comparison with genetic correlation of the same traits in UKB if the appropriate data are available.

Response:

The present study applied LD Score regression (LDSC) to perform the genetic correlation analysis, which requires only GWAS summary statistics and is not biased

by sample overlapping. Brendan Bulik-Sullivan et al, the developers of LDSC, declared that LDSC was not currently applicable to samples from recently admixed populations, which is a main limitation now. Thus, we thought it was probably not appropriate to perform the trans-ancestry genetic correlation analysis⁸.

Comment #13: Methods- Snoring: Please clarify how the snoring phenotype was assessed in the discovery CKB and replication UKB participants, as well as for the BBJ examined for genetic correlation. Were there any differences in questionnaire methodology in CKB vs. UKB and BBJ, or were the questions exactly the same?

Response:

We used a self-reported questionnaire to assess the snoring phenotype in the CKB. Participants were asked about their snoring habits: “Do you snore during sleep?” Three options were available: “Yes, frequently”, “Yes, sometimes”, or “No / Don’t know.” Participants with the first two options were classified into the snoring group. Furthermore, those who reported “frequently” snoring were defined as the habitual snoring group.

In the UKB study, baseline data were collected between 2006-2010. Snoring was a self-reported trait (Field-ID: 1210): “Does your partner or a close relative or friend complain about your snoring?” After excluding participants who answered “Don’t know” or “Prefer not to answer”, 408,000 participants were included in the GWAS, containing 37% snorers.

However, the BBJ study hasn’t mentioned a GWAS of snoring in their published paper. We have addressed this in the “Methods-Snoring” (Page 14 Line 20-22, Page 15 Line 1-2) and Supplementary Methods (Page 4 Line 4-8).

Comment #14: 9. Methods – Genome-wide association analysis: The adjustment for ten PCs of ancestry seems arbitrary. I suggest the authors examine the PCs in the population analyzed and use a systematic method to determine the appropriate number of PCs to be adjusted, as described in this reference: <https://genomebiology.biomedcentral.com/articles/10.1186/s13059-022->

02761-4

Response:

We thanked the reviewer for pointing this out. GWAS conducted in the CKB study always adjusted for the first ten principal components, which was appropriate for the current population structure among the CKB participants.

Please see our previous work⁹ (Zhu Z, Li J, Set al. A large-scale genome-wide association analysis of lung function in the Chinese population identifies novel loci and highlights shared genetic etiology with obesity. *Eur Respir J.* 2021 Oct 14;58(4):2100199. doi: 10.1183/13993003.00199-2021. PMID: 33766948).

Comment #15: 10. Methods / Results – Mendelian Randomization: Please provide a description of what approaches were used to check the assumptions of Mendelian Randomization, and summarize evidence that the assumptions were met – especially the horizontal pleiotropy assumption.

Response:

Yes, it's important to test the assumptions for the Mendelian randomization (MR) study. In the present bi-directional MR analysis, GWAS of two independent samples in East Asia, BBJ and CKB, were leveraged to avoid overlapping subjects or the possibility of population stratification.

The relevance assumption needed the genetic instrumental variables (IV) associated with the exposure. In the present MR analysis, SNPs associated with BMI at a genome-wide significant level ($P < 5 \times 10^{-8}$) were selected, and those associated with snoring at $P < 5 \times 10^{-8}$ and $P < 1 \times 10^{-5}$. *F* statistic of each SNP was calculated by $(\text{beta}/\text{standard error})^2$ to avoid the weak IV bias.

The exclusion and exchangeability assumptions needed the instrumental variables independent of the outcome given the exposure and the confounders¹⁰. Both assumptions could be unfitted if horizontal pleiotropy existed. In the present MR study, several methods were applied to reduce the horizontal pleiotropy bias. MR pleiotropy residual sum and outlier (MR-PRESSO) test¹¹ was performed for the BMI-selected SNPs (snoring SNPs were not enough to run the test), the outlier SNPs were excluded

($P < 0.05$) to eliminate the effects of pleiotropy. The reported traits of each SNP were looked up in the PhenoScanner^{12,13}, and those with traits related to the outcome in Mendelian randomization (MR) analysis were excluded. Besides, both the weighted median estimation (WME)¹⁴, and MR-Egger (ME)¹⁵ were applied as sensitivity analysis. WME provided an unbiased result when no more than 50% of invalid IVs (i.e., due to pleiotropy)¹⁴. The intercept of ME regression could indicate the degree of directional horizontal pleiotropy¹⁵. Besides, MR-Steiger tests the causal direction¹⁶. On the other hand, if heterogeneity existed across the causal effects of IVs, the horizontal pleiotropy effect might also exist. Cochran's Q test assessed the heterogeneity among IVs, and a random-effect model in IVW was applied if heterogeneity existed ($P < 0.05$). We have reported the results of testing the assumption. SNPs previously reported to be associated with the outcomes ($P < 1 \times 10^{-5}$) were excluded. The F statistic of each SNP was larger than 10, suggesting a low possibility of weak instrumental variable bias. The intercept of MR Egger regression indicated no significant horizontal pleiotropy ($P > 0.05$). Several IVW Cochran's Q tests showed the existence of heterogeneity ($P < 0.05$). Thus, the random-effect model in IVW was applied. All analyses passed the MR-Steiger test ($P < 0.001$).

We have addressed these in the Methods (Page 21, Line 16-22, Page 22, Line 1-22, Page 23 Line 1) and Results sections (Page 9, Line 16-21).

Comment #16: 11. Supp Table 1:

a. The authors state Biobank Japan (BBJ) was used to examine genetic correlation of snoring with other cardiometabolic traits, but snoring is not included in Supp Table 1. Were the GWAS summary stats of snoring available in BBJ?

Response:

We thanked the reviewer for pointing this out. However, BBJ hasn't published the GWAS of snoring or released the GWAS summary data of snoring yet.

Comment #17: b. Methods – Sensitivity analysis refers to Supp Table 1, but it seems the authors referred to the wrong Table because Supp Table 1 is not relevant to the text

in that section.

Response:

We addressed this in the Supplementary Tables as suggested by the reviewer.

Comment #18: c. Supp Table 1 seems out of place since it is not the first Supp Table referenced in the manuscript.

Response:

Done.

Comment #19: Minor:

1. Abstract: Check grammar and tense usage, as some past tense should be changed to present:

a. First sentence: “was” □ “is”

b. Last sentence: “revealed” -> “reveal”; “provided” -> “provide”

2. Introduction: Check grammar

3. Other sections of the manuscript also have awkward use of English grammar. Suggest the authors work with a native English speaker or a professional editor to improve the quality and clarity of writing.

4. Methods – Study Design: “polygenetic risk scores” -> “polygenic risk scores”

Response:

We thank the reviewer’s comments, and went over for the grammar and tense usage in our manuscript.

References

- 1 Watanabe, K., Taskesen, E., van Bochoven, A. & Posthuma, D. Functional mapping and annotation of genetic associations with FUMA. *Nature communications* **8**, 1826, doi:10.1038/s41467-017-01261-5 (2017).
- 2 Jordan, A. S., McSharry, D. G. & Malhotra, A. Adult obstructive sleep apnoea. *Lancet (London, England)* **383**, 736-747, doi:10.1016/s0140-6736(13)60734-5 (2014).
- 3 Tang, B. L. Vesicle transport through interaction with t-SNAREs 1a (Vti1a)'s roles in neurons. *Helixyon* **6**, e04600, doi:10.1016/j.helixyon.2020.e04600 (2020).

- 4 Walters, R. G. *et al.* Genotyping and population characteristics of the China Kadoorie Biobank. *Cell genomics* **3**, 100361, doi:10.1016/j.xgen.2023.100361 (2023).
- 5 Abecasis, G. R. *et al.* An integrated map of genetic variation from 1,092 human genomes. *Nature* **491**, 56-65, doi:10.1038/nature11632 (2012).
- 6 GTEx Consortium. The Genotype-Tissue Expression (GTEx) project. *Nature genetics* **45**, 580-585, doi:10.1038/ng.2653 (2013).
- 7 Ishigaki, K. *et al.* Large-scale genome-wide association study in a Japanese population identifies novel susceptibility loci across different diseases. *Nature genetics* **52**, 669-679, doi:10.1038/s41588-020-0640-3 (2020).
- 8 Bulik-Sullivan, B. *et al.* An atlas of genetic correlations across human diseases and traits. *Nature genetics* **47**, 1236-1241, doi:10.1038/ng.3406 (2015).
- 9 Zhu, Z. *et al.* A large-scale genome-wide association analysis of lung function in the Chinese population identifies novel loci and highlights shared genetic aetiology with obesity. *The European respiratory journal* **58**, doi:10.1183/13993003.00199-2021 (2021).
- 10 Didelez, V. & Sheehan, N. Mendelian randomization as an instrumental variable approach to causal inference. *Statistical methods in medical research* **16**, 309-330, doi:10.1177/0962280206077743 (2007).
- 11 Verbanck, M., Chen, C. Y., Neale, B. & Do, R. Detection of widespread horizontal pleiotropy in causal relationships inferred from Mendelian randomization between complex traits and diseases. *Nature genetics* **50**, 693-698, doi:10.1038/s41588-018-0099-7 (2018).
- 12 Staley, J. R. *et al.* PhenoScanner: a database of human genotype-phenotype associations. *Bioinformatics (Oxford, England)* **32**, 3207-3209, doi:10.1093/bioinformatics/btw373 (2016).
- 13 Kamat, M. A. *et al.* PhenoScanner V2: an expanded tool for searching human genotype-phenotype associations. *Bioinformatics (Oxford, England)* **35**, 4851-4853, doi:10.1093/bioinformatics/btz469 (2019).
- 14 Bowden, J., Davey Smith, G., Haycock, P. C. & Burgess, S. Consistent Estimation in Mendelian Randomization with Some Invalid Instruments Using a Weighted Median Estimator. *Genetic epidemiology* **40**, 304-314, doi:10.1002/gepi.21965 (2016).
- 15 Bowden, J., Davey Smith, G. & Burgess, S. Mendelian randomization with invalid instruments: effect estimation and bias detection through Egger regression. *International journal of epidemiology* **44**, 512-525, doi:10.1093/ije/dyv080 (2015).
- 16 Hemani, G., Tilling, K. & Davey Smith, G. Orienting the causal relationship between imprecisely measured traits using GWAS summary data. *PLoS genetics* **13**, e1007081, doi:10.1371/journal.pgen.1007081 (2017).

Reviewers' comments:

Reviewer #2 (Remarks to the Author):

The authors have provided reasonable responses to most of our comments from the prior review. Some additional concerns and points for clarification remain. We provide our detailed comments below. Note: The numbering system used in reference to our prior comments is based on the numbering in our original review.

Major:

- Response to our comment #2 from the prior review:

o The authors have added a descriptives table (Supp Table 1). The response document mentions the baseline survey to define geographic regions, while the main text and Supp Table 1 mention origins in the south of China. Please provide a literature reference for the baseline survey. Also, please provide a complete list of the possible geographic origin categories from the baseline survey, in order to contextualize the information provided.

o With regard to the descriptives for "non-local ancestry", we find this terminology confusing because "local ancestry" in genetic studies typically refers to admixture (e.g. among African Americans or Hispanics). Please consider rewording. Additionally, please include a citation to the prior publication that describes the authors' term "non-local ancestry" which is also used in Supp Table 1.

- Response to our prior comment #3a: The text noting the meaning of genomic regions mentions "significant independent SNPs ($P\text{-value} < 5 \times 10^{-8}$, $r^2 < 0.6$) closer than 250kb ($r^2 < 0.1$)". Please clarify why there are two different R-squared thresholds used in this sentence.

- Response to prior comment #3b:

o Please update the manuscript text further to clarify which of the genomic risk loci described were identified from primary analyses vs. sensitivity analyses (BMI-adjusted vs. sex-stratified)

o Additionally, please be consistent regarding the window size used to report novel loci. Currently, the legend of Table 1 notes "Novel loci were defined as the genomic risk loci that were more than 500 kb away from the loci identified in previous GWAS for snoring.", while the Methods text (page 17) refers to a window size of 250kb.

- Response to our prior comment #6b regarding the rationale for the significance thresholds used for replication: please incorporate in the manuscript text. Currently the rationale is provided in the response document but not in the manuscript itself.

- Response to our prior comment #6d regarding comparison of allele frequencies across groups: as the authors now refer to trans-ancestry MAF comparison, they should clarify in their comparison of CKB vs. UKB which ancestry groups were used for the MAF comparisons. This point is of importance as the UKB in particular includes participants of multiple self-reported ancestry groups.

- Response to our prior comment #7 regarding genetic correlation analysis: The authors note that they did not think it appropriate to perform trans-ancestry genetic correlation analysis due to methodological concerns. However, we suggest they could perform the genetic correlation analysis within European ancestry individuals from UKB and compare those response with their existing results from CKB and BBJ.

Response Letter

Dear Reviewers,

Manuscript ID COMMSBIO-23-1935A entitled " **Genome-wide association study of snoring and its associations with cardiometabolic traits**"

The authors would like to thank the reviewers for your valuable time and helpful comments. We appreciate the inputs you have given and that your inputs will definitely help improve our manuscript. The next section contains our point-by-point responses. We included tracked changes version of the manuscript and supplementary file.

We look forward to hearing from you regarding our submission. We would be glad to respond to any further questions and comments that you may have.

Sincerely,

Canqing Yu, MD, PhD

Associate Professor,

Department of Epidemiology & Biostatistics

Peking University Health Science Center

38 Xueyuan Road, Beijing 100191, P. R. China

Fax number: 86-10-82801620

Phone: 86-10-82801528 ext.322

Email: yucanqing@pku.edu.cn

Comment 1:

Reviewer #2 (Remarks to the Author):

The authors have provided reasonable responses to most of our comments from the prior review. Some additional concerns and points for clarification remain. We provide our detailed comments below. Note: The numbering system used in reference to our prior comments is based on the numbering in our original review.

Major - Response to our comment #2 from the prior review:

The authors have added a descriptives table (Supp Table 1). The response document mentions the baseline survey to define geographic regions, while the main text and Supp Table 1 mention origins in the south of China. Please provide a literature reference for the baseline survey. Also, please provide a complete list of the possible geographic origin categories from the baseline survey, in order to contextualize the information provided.

Response:

We thank the reviewer for pointing it out. The baseline survey of China Kadoorie Biobank (CKB) was described in the previously published papers^{1,2}. The CKB study took place in ten geographically defined regions of China, including Jiangsu, Zhejiang, Sichuan, Hunan, Guangxi, Hainan, Heilongjiang, Shandong, Gansu, and Henan (Extended Figure 1). According to the Qinling Mountains and Huai River as the north-south boundary in China, the first six study areas were in southern China, and the other four areas were in northern China.

We have added an explanation for the relationship between 10 study areas and the origins in northern/southern China in the Methods section (Page 15, Line 9-11) and the notes below Supplementary Table 1.

Extended Figure 1. Locations of the ten study areas and number of participants recruited.

Notes: The number recruited at baseline in each study area is shown in brackets. CKB study areas in northern China were colored in blue, and CKB study areas in southern China were colored in green.

Comment 2:

With regard to the descriptives for “non-local ancestry”, we find this terminology confusing because “local ancestry” in genetic studies typically refers to admixture (e.g. among African Americans or Hispanics). Please consider rewording. Additionally, please include a citation to the prior publication that describes the authors’ term “non-

local ancestry” which is also used in Supp Table 1.

Response:

We thank the reviewer pointing this out, and have modified “non-local ancestry” to “genetic outliers within each geographical origin” in the Methods (Page 17 Line 11-13), Results (Page 5 Line 14), and Supplementary Table 1.

The “non-local ancestry” was applied in our recently published paper³, in which the non-local ancestry was defined as the participants lay outside the main genotypic PC cluster, with the means of the first 11 principal components (PCs 1-11) more than three standard deviations from the area means of the PCs 1-11, and thus appeared to have non-local ancestry. We added this paper as a reference for the definition of genetic outliers within each geographical origin in the Methods (Page 17 Line 13).

Comment 3:

- Response to our prior comment #3a: The text noting the meaning of genomic regions mentions “significant independent SNPs (P-value $<5\times 10^{-8}$, $r^2<0.6$) closer than 250kb ($r^2<0.1$)”. Please clarify why there are two different R-squared thresholds used in this sentence.

Response:

Referring to the procedures of the Functional mapping and annotation platform (<https://fuma.ctglab.nl/>)⁴, a two-step clumping method based on 1000G EAS was applied in the present study.

Based on summary statistics, the first step of clumping (P-value $<5\times 10^{-8}$, $r^2<0.6$) derived independent significant SNPs (ind. sig. SNPs). SNPs with $r^2 \geq 0.6$ with any of the detected ind. sig. SNPs (called candidate SNPs) were included for functional consequences on gene functions (position and eQTL mapping). Besides, $r^2<0.6$ was used to determine the borders of the genomic risk loci.

Based on the Ind. sig. SNPs, the independent lead SNPs were defined if they were independent from each other at $r^2<0.1$. Ind. sig. SNPs dependent on each other at $r^2<0.1$ were assigned to the same genomic risk region. Also, Ind. sig. SNPs closer than 250kb were merged into one genomic risk region. Each genomic risk region was represented

by the top lead SNP which had the minimum P-value in the region, reported as the genomic risk loci in the present study.

We have addressed this in the Methods section (Page 18 Line 20-22, Page 19 Line 1-5).

Comment 4:

- Response to prior comment #3b:

o Please update the manuscript text further to clarify which of the genomic risk loci described were identified from primary analyses vs. sensitivity analyses (BMI-adjusted vs. sex-stratified)

Response:

We have separately described the genomic risk loci in each analysis in the Result section (Page 5 Line 19-23, Page 6 Line 1-21).

Comment 5:

o Additionally, please be consistent regarding the window size used to report novel loci. Currently, the legend of Table 1 notes “Novel loci were defined as the genomic risk loci that were more than 500 kb away from the loci identified in previous GWAS for snoring.”, while the Methods text (page 17) refers to a window size of 250kb.

Response:

We have unified the window size applied for the definition of novel loci as the genomic risk loci more than 500kb away from the loci identified in previous GWAS for snoring in the Methods section (Page 19 Line 7).

Comment 6:

- Response to our prior comment #6b regarding the rationale for the significance thresholds used for replication: please incorporate in the manuscript text. Currently the rationale is provided in the response document but not in the manuscript itself.

Response:

Done (Methods section: Page 21 Line 2-4, 8-10).

Comment 7:

- Response to our prior comment #6d regarding comparison of allele frequencies across groups: as the authors now refer to trans-ancestry MAF comparison, they should clarify in their comparison of CKB vs. UKB which ancestry groups were used for the MAF comparisons. This point is of importance as the UKB in particular includes participants of multiple self-reported ancestry groups.

Response:

All the CKB participants included in the current GWAS analysis were of East-Asian ancestry, which were also included for the MAF calculation. MAF of UKB participants were obtained from a published GWAS of snoring in UKB⁵, in which only participants of European ancestry were included.

We have modified this in the Methods (Page 21 Line 12-14) and Results (Page 8 Line 2-5, Page 9 Line 9-11) sections.

Comment 8:

- Response to our prior comment #7 regarding genetic correlation analysis: The authors note that they did not think it appropriate to perform trans-ancestry genetic correlation analysis due to methodological concerns. However, we suggest they could perform the genetic correlation analysis within European ancestry individuals from UKB and compare those response with their existing results from CKB and BBJ.

Response:

We have added a genetic correlation analysis conducted among the Europeans in UKB (Methods section: Page 23 Line 5-8), and the results were shown in Supplementary Table 16, Results section (Page 9 Line 9-12). We also added a comparison for the genetic correlation between East Asian and European populations in the Discussion section (Page 11 Line 19-22, Page 12 Line 1-8).

References

1. Chen, Z. *et al.* China Kadoorie Biobank of 0.5 million people: survey methods, baseline characteristics and long-term follow-up. *Int J Epidemiol* **40**, 1652-66 (2011).
2. Chen, Z. *et al.* Cohort profile: the Kadoorie Study of Chronic Disease in China (KSCDC). *Int J Epidemiol* **34**, 1243-9 (2005).
3. Walters, R.G. *et al.* Genotyping and population characteristics of the China Kadoorie Biobank. *Cell Genom* **3**, 100361 (2023).
4. Watanabe, K., Taskesen, E., van Bochoven, A. & Posthuma, D. Functional mapping and annotation of genetic associations with FUMA. *Nat Commun* **8**, 1826 (2017).
5. Campos, A.I. *et al.* Insights into the aetiology of snoring from observational and genetic investigations in the UK Biobank. *Nat Commun* **11**, 817 (2020).

REVIEWERS' COMMENTS:

Reviewer #2 (Remarks to the Author):

The authors have addressed appropriately our comments from the prior review.